# Understanding the Inner-Working of Grokking through the lens of Lottery Tickets

## Abstract

Grokking is one of the most surprising puzzles in neural network generalization: a network first reaches a memorization solution with perfect training accuracy and poor generalization, but with further training, it reaches a perfectly generalized solution. We aim to analyze the mechanism of grokking from the lottery ticket hypothesis, identifying the process to find the lottery tickets (good sparse subnetworks) as the key to describing the transitional phase between memorization and generalization. We refer to these subnetworks as "Grokking tickets", which is identified via magnitude pruning after perfect generalization. First, using "Grokking tickets", we show that the lottery tickets drastically accelerate grokking compared to the dense networks on various configurations (MLP and Transformer, and arithmetic and image classification tasks). Additionally, to verify that "Grokking tickets" are a more critical factor than weight norms, we compared the "good" subnetworks with a dense network having the same L1 and L2 norms. Results show that the subnetworks generalize faster than the controlled dense model. In further investigations, we discovered that at an appropriate pruning rate, grokking can be achieved even without weight decay. We also show that speedup does not happen when using tickets identified at the memorization solution or transition between memorization and generalization or when pruning networks at the initialization (Random pruning, Grasp, SNIP, and Synflow). The results indicate that the weight norm of network parameters is not enough to explain the process of grokking, but the importance of finding good subnetworks to describe the transition from memorization to generalization.

## 1 Introduction

The mechanism of generalization is a central question in understanding the success of neural networks. A recently observed mystery on the topic is *grokking*, which was first discovered by Power et al. (2022). Specifically, Power et al. (2022) showed the surprising phenomenon of *delayed* generalization; neural networks first reach *memorization solution* $C_{mem}$ with the perfect training accuracy but poor generalization, but further training transits the solution to perfectly *generalized solution* $C_{gen}$. Specifically, Power et al. (2022) showed the surprising phenomenon of *delayed* generalization; neural networks first reach *memorization solution* with the perfect training accuracy but poor generalization, but further training transits the solution to perfectly *generalized solution* . While the phenomenon is first observed in the modular addition task $((a+b) \bmod p$ for $a, b \in (0, \cdots, p-1))$, later it is observed in more complex datasets including MNIST (Liu et al., 2023), semantic analysis (Liu et al., 2023), n-k parity (Merrill et al., 2023), and hierarchical task (Murty et al., 2023).

*What is the underlying mechanism of the transition between memorization and generalization? What is happening during the transition period?* Prior studies provide various answers to the question, including the difficulty of learning good representations (Liu et al., 2022; Nanda et al., 2023), optimization instability (Thilak et al., 2022), and the simplicity of the generalization solution (Power et al., 2022; Liu et al., 2023; Varma et al., 2023). Among that, the most dominant explanation is a kind of last one based on the simplicity of the generalization solution, more specifically, the weight norms of network parameters $\|\boldsymbol{\theta}\|_2$, where $\boldsymbol{\theta}$ represents the parameters of neural networks. For example, the original paper (Power et al., 2022) argued that weight decay is a crucial factor in grokking. Liu et al. (2023) analyzed the loss landscapes of train and test, verifying that grokking occurs even in general deep learning tasks by modifying the weight norms of initialized parameters

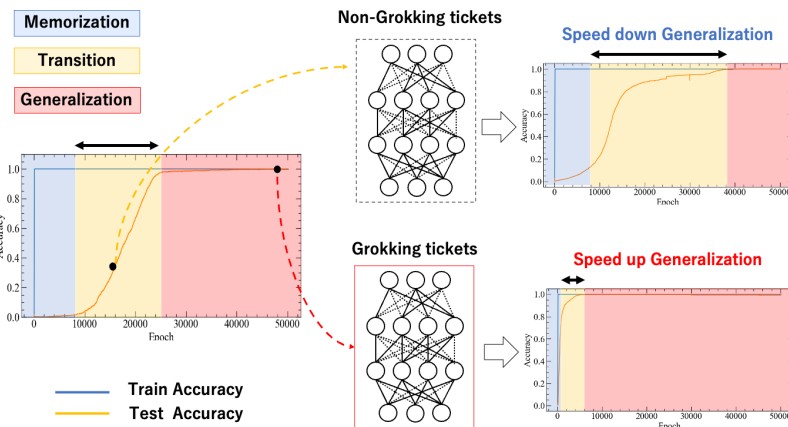

Figure 1: Illustration of grokking tickets. We show that the lottery tickets identified at the generalized solution drastically increase the speed of generalization.

$\|\boldsymbol{\theta}_0\|_2$, where $\boldsymbol{\theta}_0$ represents the initial parameters. More recently, Varma et al. (2023) hypothesized the existence of generalization circuits ($C_{gen}$) and memorization circuits ($C_{mem}$) and demonstrated that the generalization circuits could produce higher logits with smaller weight norms.

In this paper, we support that the simplicity of generalization solution is a mechanism of grokking, but further dive into deeper by connecting the inner workings of the grokking with the *lottery ticket hypothesis* (Frankle & Carbin, 2019). Specifically, we first show that the lottery tickets identified after the perfect generalization drastically accelerate grokking compared to the dense networks or lottery tickets identified before the generalization, as briefly shown in Figure 1. We denote the lottery tickets correspond to the generalized solution as **Grokking tickets** to describe the observations and show that the grokking tickets accelerate grokking on various configurations (MLP and Transformer, and an arithmetic and image classification task, in Figure 2). The results imply that the transition between the memorized and generalized solution corresponds to exploring good subnetworks, i.e., lottery tickets. Based on the observations, we further conduct a set of experiments to investigate the connection between lottery tickets, grokking, and existing observations on weight norms during the grokking process. Specifically, we conduct two types of control experiments to answer the following question: (Q1) Given the same level of weight norm, do the grokking tickets accelerate the grokking process compared to other networks? (Q2) Given the same level of network sparsity (the number of trainable parameters), do the grokking tickets accelerate the grokking process? To answer the first question, we compared the grokking tickets with the dense model with the same L1 and L2 norm, and the results show that the grokking tickets still generalize faster than the controlled dense model. We also found that, with the precise selection of the pruning rato, the grokking tickets do not require weight decay to accelerate grokking further, suggesting that weight decay is crucial for uncovering the lottery tickets but becomes redundant after their discovery. For the second question, we compared the grokking tickets to differently identified subnetworks, including several pruning at initialization methods (random pruning, Grasp (Wang et al., 2020), SNIP (Lee et al., 2019), and Synflow (Tanaka et al., 2020)) and lottery tickets identified at the memorization solution. The results show that poor selection of the subnetworks hurts the generalization, suggesting that grokking tickets hold good properties beyond just a sparsity. These results validate our hypothesis that exploring the good subnetworks is an inner-working during the transition between memorization and generalization, rather than just reducing the weights norms as prior research suggested.

## 2 NOTATION AND PRELIMINARIES

### 2.1 GROKKING

This paper focuses on grokking on the classification tasks commonly used in grokking phenomenon studies (Power et al., 2022; Nanda et al., 2023; Liu et al., 2022; 2023). Assume we have training

datasets $\mathbb{S}_{train}$ and test datasets $\mathbb{S}_{test}$, and train a neural network $f(\boldsymbol{x}; \boldsymbol{\theta})$ where $\boldsymbol{x}$ is an input and $\boldsymbol{\theta}$ are weight parameters of networks. Specifically, they train the network using stochastic gradient decent over classification loss and weight decay (L2 norm of weights $\|\boldsymbol{\theta}\|_2$):

$$\arg\min_{\boldsymbol{\theta}} \mathbb{E}[L(f(\boldsymbol{x}; \boldsymbol{\theta}), y) + \frac{\alpha}{2}\|\boldsymbol{\theta}\|_2],$$

where $y \in \{1, ..., k\}$ is a class labels correspond to the inputs $\boldsymbol{x}$, and $\alpha$ is a weighting parameters. Power et al. (2022) shows that the optimization process finds a memorization solution $C_{mem}$ at the earlier training but suddenly transits to the perfectly generalized solution $C_{gen}$ on certain types of tasks and network architectures. Here, $C_{mem}$ refers to the solution with almost perfect training performance but poor test performance, and $C_{gen}$ is the almost perfectly generalized solution. Specifically, the training accuracy reaches $g\%$ after $t_{mem}$ iterations ($g$ is the 99% in the original paper), but test accuracy reaches it after $t_{gen} >> t_{mem}$ iterations (e.g., $t_{gen}$ is $10^2 \sim 10^3$ and $t_{mem}$ is $10^6$ in the original paper). This is a surprising finding since the loss of the network should be already saturated at the $t_{mem}$, and thus, the underlying mechanism during the transition between $C_{mem}$ and $C_{gen}$ (denoted as *transition phase*) is not trivial. Throughout the paper, we denote the weights at iteration $t$ as $\boldsymbol{\theta}_t$, and $\boldsymbol{\theta}_{gen}$ and $\boldsymbol{\theta}_{mem}$ for the weights at the $\boldsymbol{\theta}_{t_{gen}}$ and $\boldsymbol{\theta}_{t_{mem}}$ respectively, with slightly abuse the notion. We also use $t_{gen}$ or $t_{gen} - t_{mem}$ as the measurements of the grokking speed, i.e., how fast the networks transit to the generalization solution from the memorization solution.

### 2.2 LOTTERY TICKETS, MAGNITUDE PRUNING, AND PRUNING AT INITIALIZATION

In this paper, we couple the grokking phenomenon with the other interesting observations in training over-parameterized networks, *lottery tickets hypothesis* (Frankle & Carbin, 2019). Informally, the lottery tickets hypothesis states that randomly initialized over-parameterized networks include sparse subnetworks that reach good performance after train, and the existence of the subnetworks is key to achieving good generalization in deep neural networks.

**Definition 2.1** (Lottery Ticket Hypothesis). Let $f(\boldsymbol{x}; \boldsymbol{\theta}_0)$ be a dense network whose weights are randomly initialized ($\boldsymbol{\theta}_0 \sim \mathcal{D}_{\boldsymbol{\theta}}$). After training $f(\boldsymbol{x}; \boldsymbol{\theta}_0)$ on a training dataset for $j$ iterations, it reaches a minimum validation loss with a test accuracy $a$. Then, there exists a mask $\boldsymbol{m} \in \{0, 1\}^{|\boldsymbol{\theta}_0|}$ such that $f(\boldsymbol{x}; \boldsymbol{\theta}_0 \odot \boldsymbol{m})$ reaches a test accuracy $a'$ after training for $j'$ iterations on the same setting, where $a' \geq a$, $j' \leq j$, and $\|\boldsymbol{m}\|_0 \ll |\boldsymbol{\theta}|$.

Here, $\|\boldsymbol{m}\|_0$ is the number of non-zero elements of the mask $\boldsymbol{m}$. This paper refers to the mask itself or the sparse subnetworks $f(\boldsymbol{x}; \boldsymbol{m} \odot \boldsymbol{\theta}_0)$ as *lottery tickets (LT)*. Usually, LT is identified by using some measurements to determine each weight's importance. A dominantly used measurement is the magnitude of the weights after training. Specifically, after training neural networks with certain iterations and obtaining $\boldsymbol{\theta}_T$, select the top $k$ weights with the largest absolute values. The process is called the *magnitude pruning*. In the control experiments, we also compare the grokking tickets with several *pruning at initialization (PaI)* methods (Grasp (Wang et al., 2020), SNIP (Lee et al., 2019), and Synflow (Tanaka et al., 2020)). Unlike the above-mentioned magnitude pruning, these methods prune the networks at the initialization. Each method uses different measurements to determine the importance of weights. While it is computationally efficient since it does not require any training to determine the lottery tickets, the effectiveness of these tickets is under suspicion. For example, Frankle et al. (2021) shows that randomly chaining the weights still performs similarly, meaning these methods only determine the reasonable pruning rate but do not find good subnetworks. We also show that PaI cannot find any grokking tickets, and thus, it does not accelerate the grokking process.

## 3 EXPERIMENTAL SETUP

As mentioned earlier, we investigate the mechanism of grokking from the lens of the lottery tickets, i.e., the exploration process to find the good subnetworks. This section defines the notion of the *grokking tickets* and how we find them. We then show that grokking tickets accelerate the grokking process using two types of classification tasks (Modular addition and MNIST classification) and architectures (MLP and Transformer).

### 3.1 Identifying Grokking Tickets and Baselines

This paper refers to the *grokking tickets* as lottery tickets identified at the generalization solution $C_{gen}$. Figure 1 illustrates the grokking tickets. Specifically, we identify the grokking tickets as follows:

1. Randomly initialize a neural network $f(.; \boldsymbol{\theta}_0)$.

2. Train the network until full generalization (after grokking), arriving at parameters $\boldsymbol{\theta}_{gen}$.

3. Create a mask vector $\boldsymbol{m}$ by pruning $k\%$ of the parameters in $\boldsymbol{\theta}_{gen}$ according to the score $S(w_i)$, where $k$ is a pruning rato.

4. Reset the remaining parameters to their values in $\boldsymbol{\theta}_0$, and obtain the grokking ticket $f(\boldsymbol{x}; \boldsymbol{m} \odot \boldsymbol{\theta}_0)$

Specifically, we use the magnitude as a score function $S(w_i) = |w_i|$ Note that most prior studies in LT use an iterative version of the above process, i.e., gradually increasing the pruning rato by repeating the above process. The process is known to be vital to finding good subnetworks, especially when we want to use extremely high sparsity. However, we use the one-shot version of the magnitude pruning in the experiments because our focus is not to identify the best lottery tickets and does not require extremely high sparsity, as shown in the experiments section.

In the experiments, we compare the grokking speed of **Grokking tickets** $f(\boldsymbol{x}; \boldsymbol{m} \odot \boldsymbol{\theta}_0)$ with the following models.

**(1)** **Base model** that train the original dense model $f(.; \boldsymbol{\theta}_0)$ (subsection 4.1).

**(2)** **Non-grokking tickets** whose subnetwork is identified before grokking (subsection 4.2).

**(3)** **Controlled dense model** that has weights with the same L1 or L2 norm as grokking tickets (subsection 5.1).

**(4)** **PaI tickets** whose subnetwork is identified at initialization (subsection 5.2).

Without mentioning otherwise, we use the same hyper-parameters to train these models, e.g., the same weight decay and optimization algorithm.

### 3.2 Tasks and Architectures

Most prior studies in grokking focus on specific types of the task (Merrill et al., 2023) and architectures (MLP or Transformers (Nanda et al., 2023)). To show the universality of the analysis, we tested grokking tickets on two types of tasks (Modular addition and MNIST) and two architectures (MLP and Transformer).

#### 3.2.1 Modular addition

Similar to the original paper (Power et al., 2022), we constructed a dataset of equations of the form: $(a + b)\%p = c$.

The task is to predict $c$ given a pair of $a$ and $b$. We use the following detailed configurations in our setting: $p = 67$, $0 \leq a, b, c < p$. Considering all possible pairs of $a$ and $b$, therefore dataset size is 4489. We split the dataset into training (40%) and test (60%). In the experiments on the modular addition, we used two architectures: (1) MLP and (2) Transformer. In both architectures, we encode the $a$ and $b$ into one hot vector. For both architectures, the cross entropy loss is used.

**MLP** Following (Liu et al., 2022), we design the MLP as follows. Firstly, we map the one hot encoding of $\boldsymbol{a}$, $\boldsymbol{b}$ with the embedding weights $W_{emb}$: $E_a = W_{emb}\boldsymbol{a}, E_b = W_{emb}\boldsymbol{b}$. We then feed the embedding $E_a$ and $E_b$ into MLP as follow: $\text{softmax}(\sigma((E_a + E_b)W_{in})W_{out}W_{unemb})$, where $W_{emb}, W_{in}, W_{out}, W_{unemb}$ are the trainable parameters, and $\sigma$ is an activation function (ReLU Nair & Hinton (2010)). The dimension of the embedding space is 500, and $W_{in}$ projects into 48-dimensional neurons.

**Transformer**  Following (Nanda et al., 2023), we use the 1-layer ReLU transformer, which only has one transformer block with one attention head. We set both the embedding size and the dimension of hidden units in feed-forward networks to 500. We omit Layer Norm (see the details in subsection A.2).

### 3.2.2  MNIST CLASSIFICATION

Following (Liu et al., 2023; Varma et al., 2023) we split the dataset into 1k training examples and 60k testing examples. By reducing the amount of training data, we make it easier to memorize the training data and induce the phenomenon of grokking. In MNIST classification, we use a 4-layer MLP with ReLU activation function. The dimension of each layer is (784, 200, 200, 10). Referencing Liu et al. (2023), we are setting the $\kappa$ (defined in 3.3) to 8.0 at initialization and the strength of the weight decay $\alpha$ to 9.0. Referring to Liu et al. (2023), in the case of MNIST class classification, mean squared error (MSE) was used instead of cross-entropy. (see the details in subsection A.1)

### 3.3  OPTIMIZATION

Following (Nanda et al., 2023), we used AdamW optimizer (Loshchilov & Hutter, 2019) with learning rate $10^{-3}$, the weighting of weight decay $\alpha = 1.0$, $\beta_1 = 0.9$, $\beta_2 = 0.98$. We initialize weights as follows : $\boldsymbol{\theta}_0 \sim \mathcal{N}(0, \kappa/\sqrt{d_{in}})$, where $d_{in}$ represents the dimensionality of the layer preceding each weight. If nothing is specified, assume $\kappa = 1$.

## 4  EXPERIMENTS: LOTTERY TICKETS ACCELERATES GROKKING

### 4.1  GROKKING TICKETS VS. BASE MODEL

Figure 2 compares the grokking speed of the base model and the grokking tickets. The top row shows the transition of test loss, and the bottom row shows the test accuracy. Each column corresponds to different configurations of the task (modular addition and MNIST) and the architecture (MLP and Transformer). For grokking tickets, we set the pruning rato to 0.6 and apply the magnitude pruning after 30k iterations. We repeat the experiments with three different seeds and report the average and the standard deviation. The results clearly show the benefits of the grokking tickets, i.e., it drastically speeds up the generalization. For example, on the modular addition using MLP, the grokking tickets generalize around at $5k$ epoch, while the base model takes more than $25k$ iterations to achieve comparable test performance.. Similar results are also observed in Transformer (middle), datasets (right), and performance measurements (top).

### 4.2  GROKING TICKETS VS. NON-GROKKING TICKETS

We also compare the grokking speed of grokking and non-grokking tickets. Specifically, we first identify the lottery tickets at a different iterations: 2k, 5k 10k, 16k and 28k. We use the same pruning rate (0.6) and the same procedure (global magnitude pruning) for all tickets. For this experiment, we employed the modular addition task and used an MLP architecture. In the setup, only 28k corresponds to the grokking tickets, 2k corresponds to the memorization phase, and 10k to 16k corresponds to the transition phase (see Figure 1). The results indicate that the test accuracy (represented by a solid line) of grokking tickets is much higher than other tickets, while all tickets achieve the same level of training accuracy at the beginning of training. Additionally, the right panel of Figure 3 illustrates the relationship between the test accuracy and the epochs in which the tickets are obtained. We observe a gradual increase in test accuracy in relation to the epoch at which the lottery tickets were identified, suggesting a progressive discovery of the generalized lottery ticket. In other words, during the transition phase, the neural network gradually transforms towards a generalized solution and removes memorization components. This result is consistent with the analysis of Nanda et al. (2023), which splits training into three phases: memorization, circuit formation, and clean up.

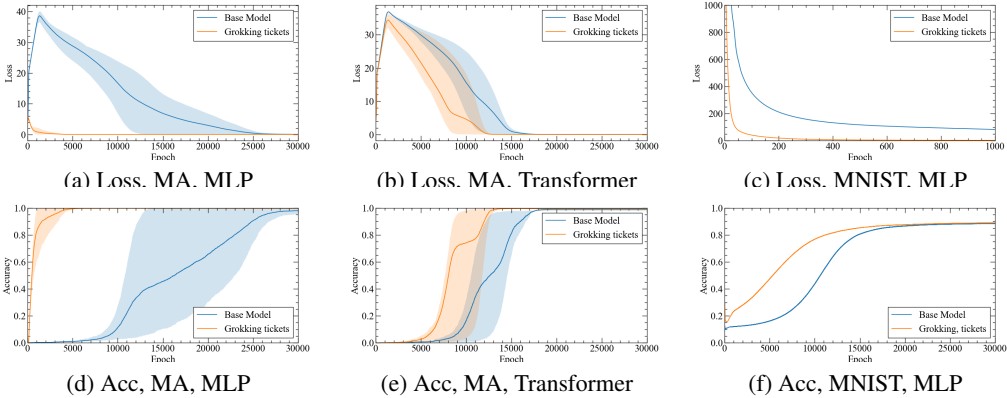

Figure 2: Comparison of base model (blue) and grokking tickets (orange). The top row shows the transition of train and test loss, and the bottom row shows the train and test accuracy. Each column corresponds to the different configurations of the task (modular addition and MNIST) and the architecture (MLP and Transformer). The dashed line represents the results of the training data.

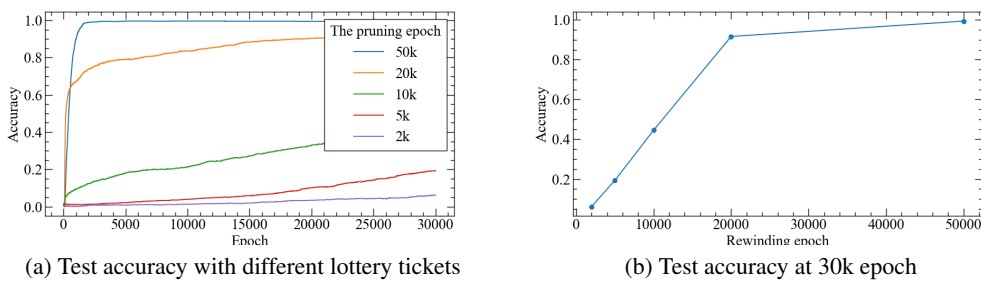

Figure 3: Comparing the grokking speed of grokking tickets (50k) vs. non-grokking tickets (2k, 5k, 10k, 16k). The number corresponds to the epoch where we identify the lottery tickets. The dashed line represents the results of the training data.

## 4.3 THE EFFECTS OF PRUNING RATE ON GROKKING TICKETS

In subsection 4.1, we only tested when the pruning rate was 0.6. The left panel of Figure 4 depicts the generalization speed across various pruning rates, while the right panel demonstrates the correlation between pruning rate and grokking speed. We use the grokking tickets in this setup. The task is modular addition and MLP architecture. A pruning rate of 0.0 represents the base model because no weights are pruned and the model is rewound to its initial weights. We can make the following observations: (1) Most pruning rato (0.1, 0.3, 0.5 and 0.7) accelerate the grokking, indicating that the above observation does not heavily depend on the selection of the pruning rato. When the pruning rate is 0.5, the speed of grokking is the fastest, according to the right figure. (2) At the same time, with a higher pruning rate, such as 0.9, the grokking phenomenon is not observed, which might be due to the lack of capacity. Even with a low pruning rate, grokking occurs, but it doesn't happen when the rate is too high. This result implies that the model still has redundant weights, which are likely removed by weight decay.

# 5 CONTROL EXPERIMENTS: GROKKING TICKETS, WEIGHT NORM AND NETWORK SPARSITY

In section 5, we show that the grokking tickets clearly accelerate the generalization speed, suggesting that the inner workings of the transition phase somehow correspond to the exploration of the good

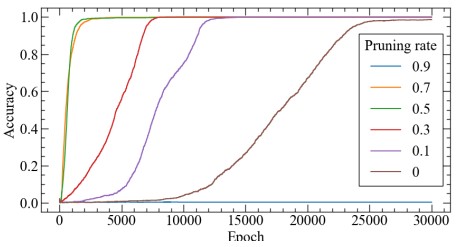 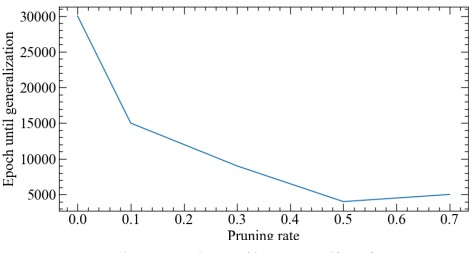

(a) Test accuracy with different pruning rate      (b) Epoch until generalization

Figure 4: The effect of pruning rate on the grokking speed of the grokking tickets. Prune rate $0.0$ corresponds to the base model. The dashed line represents the results of the training data.

subnetworks (lottery tickets). In this section, we further conduct experiments to investigate the connection between lottery tickets, grokking, and existing observations on weight norms during the grokking process. Specifically, we conduct two types of control experiments to answer the following question:

Q1. Given the same level of weight norm, do the grokking tickets accelerate the grokking process compared to other networks?

Q2. Given the same level of network sparsity (the number of trainable parameters), do the grokking tickets accelerate the grokking process?

### 5.1 CONTROL EXPERIMENTS 1: CONTROLLING WEIGHT NORM

In section 4, we show that grokking tickets accelerate grokking. However, the relationship between L2 norm and grokking (Liu et al., 2023) and the importance of weight decay in inducing grokking (Power et al., 2022) have already been mentioned in previous studies. Therefore, the reader may wonder if the grokking tickets contribute only to reducing the L2 norm, i.e., the structure itself does not have any critical role. In this section, we investigate whether the L2 norm or the grokking tickets are more plausible explanations of the grokking mechanism. To answer Q1, we first control the weight norm of the grokking tickets and the dense network with the following procedure:

1. Obtain grokking tickets using a method similar to subsection 4.1

2. Get weight $L_p$ norm ratio $r_p = \frac{\|\boldsymbol{\theta}_0 \odot \boldsymbol{m}\|_p}{\|\boldsymbol{\theta}_0\|_p}$

3. Create weights $\boldsymbol{\theta}_0 \cdot r_p$ with the same $L_p$ norm as the grokking tickets.

Therefore, the distribution that $\boldsymbol{\theta}_0 \cdot r_p$ is modified as follows : $\boldsymbol{\theta}_0 \sim \mathcal{N}(0, (\kappa \cdot r_p)/\sqrt{d_{in}})$ Figure 5 of the left show that, despite having equal norms at initialization, the model initialized with grokking tickets converges to a generalization solution faster than any other model. This indicates that the transition between memorization and generalization corresponds to the process of finding good lottery tickets rather than simply reducing weight norms, as previously assumed by some studies. Additionally, as Figure 5 of the middle and right panel shows, the L1 norm of the grokking tickets is smaller compared to others, while the L2 norm is larger. From this, it is evident that the sparsity of grokking tickets is high, and it is also apparent that the norm of some weights has increased.

**Critical pruning rate: Grokking without weight decay**    Prior research (Power et al., 2022) has emphasized the importance of weight decay in grokking. However, Figure 5 results indicate the inner-working of the transition between memorization and generalization corresponds to the exploration of the good subnetworks. If this is correct, is it possible to induce grokking without weight decay during training when using the grokking tickets? To answer the question, we first explore the *critical pruning ratio*, which is the maximum pruning rate that can induce grokking. Thus, we gradually increased the pruning rate in increments of 0.01 from 0.8 and found that the $k = 0.81$ is the critical pruning ratio. We then compare the behavior of the grokking tickets with the critical pruning

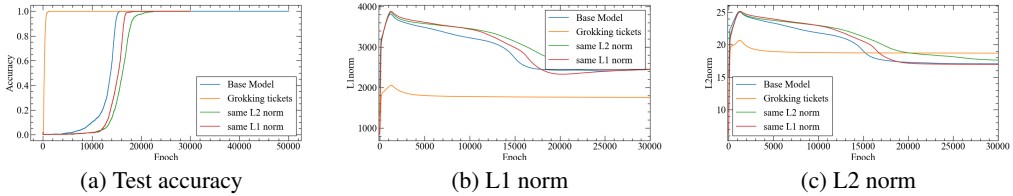

(a) Test accuracy       (b) L1 norm       (c) L2 norm

Figure 5: The respective Test accuracy (left), L1 norm (middle), L2 norm (right) of models initialized with equal L2 norm and L1 norm as with grokking tickets.

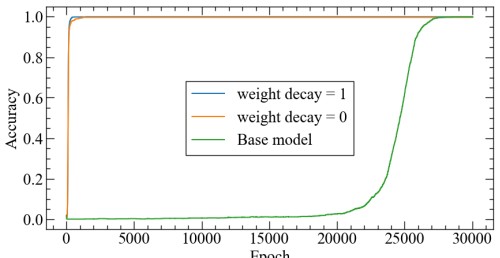

Figure 6: The effect of pruning rate on test accuracy without weight decay(left). Test accuracy of grokking tickets with critical pruning rate(0.81) with and without weight decay(right).

rato with ($\alpha = 1.0$) and without weight decay ($\alpha = 0.0$). Figure 6 show the results of the experiments. For the baseline, we also plot the test accuracy when using the base model. As shown in the figure, the test accuracy reaches perfect generalization almost similarly with and without weight decay. The results show that the grokking tickets with the critical pruning ratio do not require any weight decay during the optimization, indicating that the weight decay pushes the networks to find good lottery tickets during the transition between memorization and generalization, rather than just decreasing the weight norm of the networks.

## 5.2 CONTROL EXPERIMENTS 2: CONTROLLING NETWORK SPARSITY

For the second question Q2, we compared the grokking tickets with the subnetworks identified differently. Specifically, we tested three well-known pruning at initialization methods (Grasp (Wang et al., 2020), SNIP (Lee et al., 2019), and Synflow (Tanaka et al., 2020)) and random pruning as baseline methods. For details on each of the pruning methods, refer to the section Appendix B. Figure 7 compare the transition of the test accuracy of these PaI methods and the grokking tickets. The results show that all PaI methods perform worse than the base model or, in some cases, perform worse than the random pruning. The results show that poor selection of the subnetworks hurts the generalization, suggesting that grokking tickets hold good properties beyond just a sparsity.

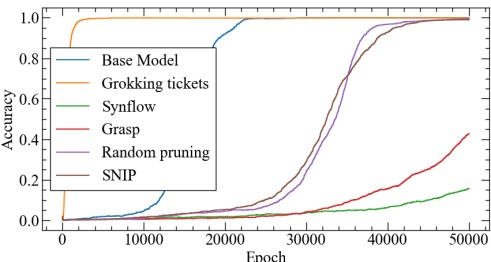

Figure 7: Comparing test accuracy of the different pruning methods.

## 6  DISCUSSION AND RELATED WORKS

In this section, we connect our results with prior explanations of the mechanism of grokking and relevant related works.

### 6.1  WEIGHT NORM

Power et al. (2022) argued that for grokking, weight decay and the ratio of training data to test data are crucial. Liu et al. (2023) analyzed the loss landscape of train and test, verifying that grokking occurs even in general deep learning tasks from the perspective of weight norm. Varma et al. (2023) hypothesized the existence of generalization circuits and memorization circuits and demonstrated that the generalization circuits can produce higher logits with small norm parameters, indicating higher efficiency. Our study is well aligned with these studies in the sense that these studies assume the simplicity of the solution is key to achieving the generalized solution. However, our studies further suggest that the sparsity of the networks is important in the process of grokking, i.e., the transition of memorization to generalization corresponds to the exploration of good subnetworks.

### 6.2  REPRESENTATION LEARNING

Liu et al. (2022) analyzed that the dynamics of representation learning and the dependency on the training set size have been predicted, demonstrating that the origin of grokking lies in learning structured representations. Nanda et al. (2023) comprehends the underlying training dynamics in grokking and can divide the training into three phases: memorization of training data, circuit formation, and cleanup. The relationship between representation learning and the discovery of lottery tickets is still not well understood. In Appendix G, following to Pearce et al. (2023), we investigated the periodicity of each neuron by plotting input-side weight and output-side weight of each neuron. The results demonstrate that grokking tickets contribute to the acquisition of a good representation.

### 6.3  SPARSITY AND EFFICIENCY

Merrill et al. (2023) argued that the grokking phase transition corresponds to the emergence of a sparse subnetwork that dominates model predictions. They empirically studied the internal structure of networks undergoing grokking on n-k parity. The objectives of this study are similar to those of our research. However, we claim that it's not just sparsity that's important, but the acquisition of good structures is crucial. In Appendix I,J, to prove our claim, we have conducted experiments with equal learning parameters and analyzed the impact of dead neurons. Additionally, Merrill et al. (2023) has only conducted experiments in n-k parity where the sparsity of optimal network structures is inherently apparent on MLP. On the other hand, our analysis method involves pruning of weights (lottery tickets), not neurons, and it applies to a diverse range of tasks and architectures.

### 6.4  LOTTERY TICKETS HYPOTHESIS

As mentioned in section 3, the method of discovering Grokking tickets is the same as that for finding original Lottery ticket (Frankle & Carbin, 2019). The novelty of this research in relation to the Lottery Tickets hypothesis is as follows: (1) Lottery Tickets as an explanation for the Grokking phenomenon. (2) Lottery Tickets that acquire good structures for generalization, not just a reduction in parameters. In Appendix G,H,I,J, we conducted experiments and analyses on the relationship between these good structures and Grokking tickets, as well as on how they are acquired.

## 7  CONCLUSION

In this paper, we analyzed grokking from the perspective of lottery tickets. Firstly, with grokking tickets, we show that the lottery tickets dramatically accelerate grokking. In addition, we compared the grokking tickets with dense models with the same norm as grokking tickets, and the results show that the grokking tickets still generalize faster than the controlled dense model. We also compared the grokking tickets with the subnetworks identified differently, and the results show that poor selection of the subnetworks hurts the generalization, suggesting that grokking tickets hold good properties beyond just a sparsity.

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
