# A  MODEL DETAIL

## A.1  MLP FOR MNIST

As mentioned section 3, we use 4-layer MLP for the MNIST classification. The difference from regular classification is that we are using Mean Squared Error (MSE) for the loss. We adopted this setting following prior research (Liu et al., 2023). In (Liu et al., 2023), it was confirmed in the Appendix that grokking occurred without any problems even when trying with cross-entropy.

## A.2  TRANSFORMER FOR MODULAR ADDITION

Similar to Nanda et al. (2023), we use 1-layer transformer in all experiments. We use single-head attention and omit layer norm.

We denote our hyperparameters as follows: $d_{vocab} = 67$ is the size of the input and output spaces, the same as $p$ in section 3, $d_{emb} = 500$ is the embedding size, $d_{mlp} = 128$ is width of the mlp layer.

We denote the parameters as follows: $W_E$ (embedding layer), $W_{pos}$ (positional embedding), $W_Q$ (query) ,$W_K$ (keys), $W_V$ (values), $W_O$ (attention output), $W_{in}$ and $b_{in}$ (the first layer weights and bias of the MLP ), $W_{out}$ and $b_{out}$ (the second layer weights of the MLP) and $W_U$ (unembedding layer).

We describe the process of obtaining the logits for a single-layer model. Note that loss is only calculated from the logits on the final token. We denote the token in position $i$ at $l$ layer as $x_i^{(l)}$. Note that $i$ is 0 or 1, because the number of input tokens is 2 and $x_i^{(0)}$ is one-hot vector. We denote the attention scores as $A$. We denote the triangular matrix with negative infinite elements as $M$, which is used for causal attention. The logits are calculated via the following equation:

$$x_i^{(1)} = W_E x_i^0 + W_{pos} x_i^{(0)}$$
$$A = \text{softmax}(\text{x}^{(1)\text{T}} \text{W}_\text{K}^\text{T} \text{W}_\text{Q} \text{x}^{(1)} - \text{M}) + x^{(1)}$$
$$x^{(2)} = W_0 W_V (x^{(1)} A) + x^{(1)}$$
$$x^{(3)} = W_{out} \text{ReLU}(W_{in} x^{(2)} + b_{in}) + b_{out} + x^{(2)}$$
$$\text{Logits} = \text{softmax}(W_U x^{(3)})$$

# B  PRUNING AT INITIALIZATION METHODS

Currently, the methodologies of pruning neural networks (NN) at initialization (such as SNIP, GraSP, SynFlow) still exhibit a gap when compared to methods that use post-training information for pruning (like Lottery Ticket). Nonetheless, this area is experiencing a surge in research activity.

The basic flow of the pruning at initialization is as follows:

1. Randomly initialize a neural network $f(\boldsymbol{x}; \boldsymbol{\theta}_0)$.
2. Prune $p\%$ of the parameters in $\boldsymbol{\theta}_0$ according to the scores $S(\boldsymbol{\theta})$, creating a mask m .
3. Train the network from $\boldsymbol{\theta}_0 \odot \boldsymbol{m}$.

According to Tanaka et al. (2020), research on pruning at initialization boils down to the methodology of determining the score in the above process 2, which can be uniformly described as follows:

$$S(\boldsymbol{\theta}) = \frac{\partial R}{\partial \boldsymbol{\theta}} \odot \boldsymbol{\theta}$$

When the $R$ is the training loss $\mathbb{L}$, the resulting synaptic saliency metric is equivalent to $|\frac{\partial L}{\partial \boldsymbol{\theta}} \odot \boldsymbol{\theta}|$ used in SNIP (Lee et al., 2019). $-(H\frac{\partial L}{\partial \boldsymbol{\theta}}) \odot \boldsymbol{\theta}$ use in Grasp (Wang et al., 2020).Tanaka et al. (2020) proposed synflow algorithm $R_{SF} = 1^T (\prod_{l=1}^{L} |\boldsymbol{\theta}^{|l|}|)1$. In section 5.2, all initial values were experimented with using the same weights and the same pruning rate.

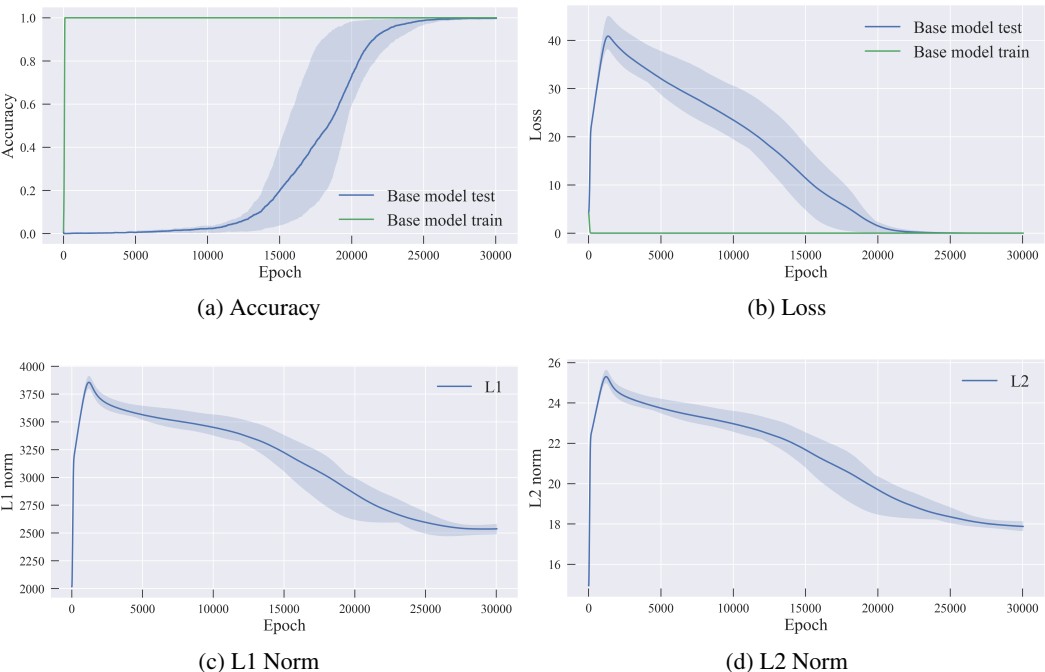

Figure 8: Typical grokking on modular addition task, MLP architecture

## C WIGHT NORM ANALYSIS ON TYPICAL GROKKING

Power et al. (2022) discovered the grokking phenomenon on small arithmetic datasets and showed weight decay is necessary for grokking. In this section, we also analyze the base model used in this paper from the perspective weight norm, similar to the prior research (Power et al. (2022), Liu et al. (2023)).

Figure 8 (a) (b) shows the dynamics of accuracy and loss on modular addition task, MLP architecture. While training accuracy immediately saturates, test accuracy achieves 100% at around 25k epochs. Of course, similar occurrences are observed with the loss as well. In this case, from section 2, the memorization phase is 0-10k epoch, the transition phase is 10k-25k epoch, and the generalization phase is 25k-30k epoch.

Figure 8 (c) (d) shows the changes in L1 and L2 norm of weight until arriving at generaliztion solution. Before 5k epoch, especially at the beginning of memorization, the weight norm dramatically increased. Subsequently, the norm decreases to 25k epoch (generalization). After generalization, as mentioned in subsection 2.1, the training loss and weight decay loss are balanced, and the norm remains unchanged.

Figure 9 is same experiments with transformer as MLP(Figure 8). It can be observed in Figure 9 as well, similar to Figure 8.

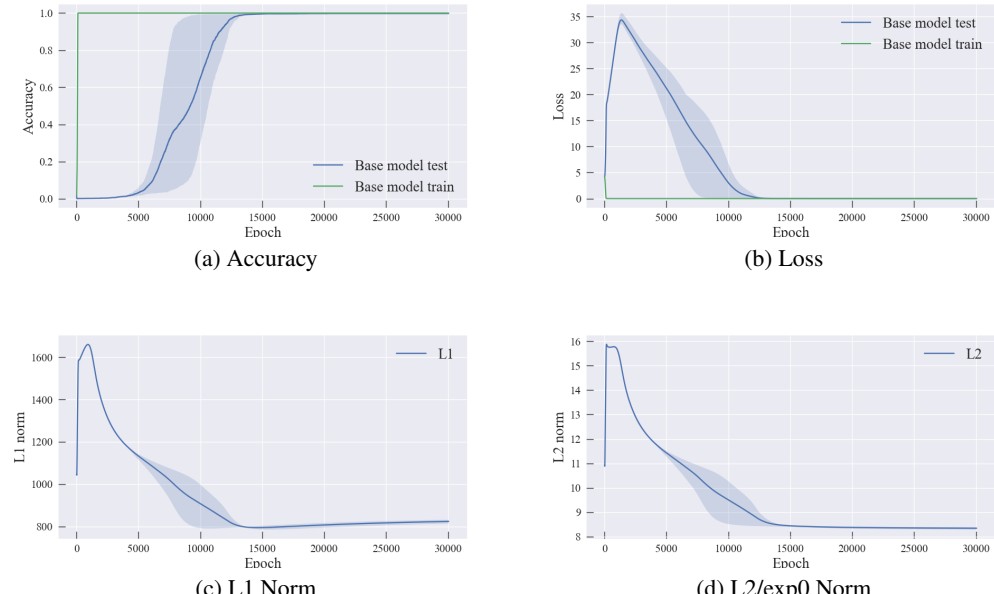

Figure 9: Typical grokking on modular addition task, Transformer architecture

## D  TRANSFORMER EXPERIMENTS

Similar to MLP experiments in section 4, we conduct experiments single-head transformer architecture.

### D.1  GROKKING TICKETS VS. NON-GROKKING TICKETS

We compare the grokking speed of grokking and non-grokking tickets. Specifically, we first identified lottery tickets at a different iteration (2k, 6k, 10k, 16k, 28k). We use the same pruning rate (0.6) and the same procedure (global magnitude pruning) for all tickets. We use the base model used in subsection 4.1 as the model to be pruned. This procedure is the same as the experiments on MLP subsection 4.2.

In the case of a transformer, especially in the attention layer, we do not prune the connection between layers but prune $W_Q$ (query matrix), $W_K$ (key matrix), which constructs the attention matrix. Therefore, it differs from pruning in the case of MLP. (see detail in subsection 6.1)

The left panel of Figure 10 shows grokking speed of grokking tickets and non-grokking tickets. Lottery tickets identified in the transition phase (10k) can be generalized a little bit after grokking tickets (16k, 28k). This is different from the result of MLP (Figure 3) that might arise from the way of pruning. On the other hand, the right panel of Figure 10 is consistent with the result of MLP experiment (Figure 3. This implies that the test accuracy gradually increases with respect to the epoch to obtain the lottery tickets in the transformer as well.

### D.2  THE EFFECTS OF PRUNING RATE ON GROKKING TICKETS

We compare the generalization speed with different pruning ratios on the transformer in Figure 11. The similarities compared to MLP experiment(Figure 4 is that with a large pruning rate, grokking is hard to occur or it takes more time to reach a generalization solution. On the other hand, the difference in results compared to MLP experiments is that it does not reach a complete generalization solution without the appropriate pruning rate.

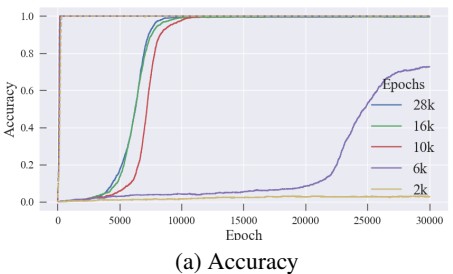 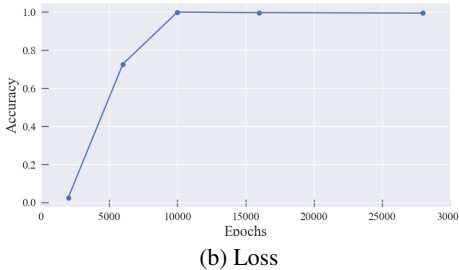

    (a) Accuracy                  (b) Loss

Figure 10: Comparing the grokking speed of grokking tickets (28k, 16k) vs. non grokking tickets(2k, 6k 10k). The number corresponds to the epoch where we identify the lottery tickets.

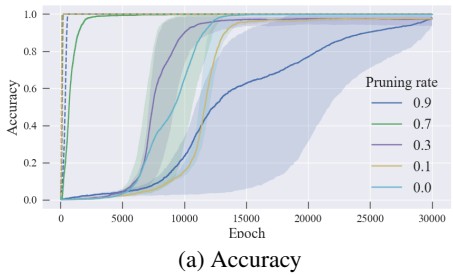 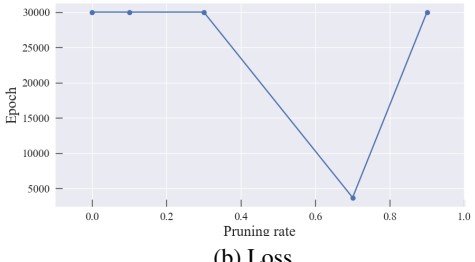

    (a) Accuracy                  (b) Loss

Figure 11: Comparing the grokking speed of grokking tickets (28k) vs. non

## D.3 CONTROLLING WEIGHT NORM

Similar to subsection 5.1, we investigate whether the weight norm or the grokking tickets are a more plausible explanation of the grokking mechanism using Transformer. In Figure 12, similarly to MLP experiment (Figure 5), it is observed that sparsity (good structure), rather than weight norm, is crucial for grokking.

## D.4 GROKKING WITHOUT WEIGHT DECAY

Similar to section 5.1, we explore the critical pruning rate using a transformer. Figure 13 shows the effect of pruning rate on the test accuracy of grokking tickets without weight decay. At a low pruning rate (0.1, 0.2, 0.3, 0.4, 0.5), the test accuracy is almost the same as random accuracy $\left(\frac{1}{67} \sim 0.014\right)$ because there are many redundant edges in the model. On the other hand, at a large pruning rate (0.9), the test accuracy is also low because even the good edges are pruned. Unlike MLP experiment (section 5.1), the critical pruning rate that fully generalizes without weight decay is not found. This might be attributable to the structural differences between MLP and Transformer (see detail

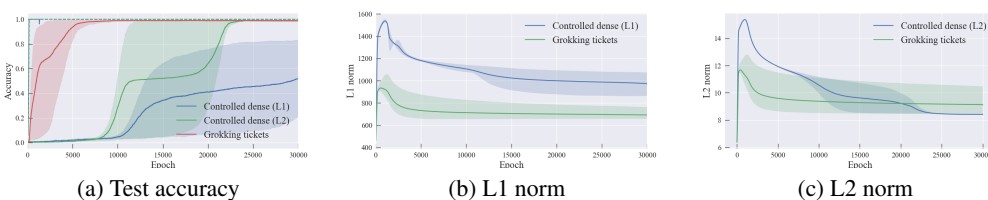

    (a) Test accuracy           (b) L1 norm             (c) L2 norm

Figure 12: The respective Test accuracy(left), L2 norm (middle), L1 norm (right) of models initialized with equal L2 norm and L1 norm as with grokking tickets.

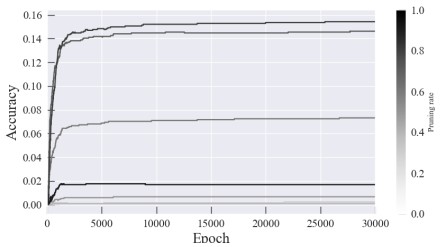 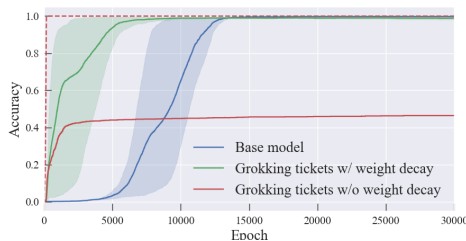

(a) Test accuracy with different pruning rate  (b) Test accuracy with the critical pruning rato

Figure 13: The effect of pruning rate on the test accuracy of grokking tickets without weight decay.

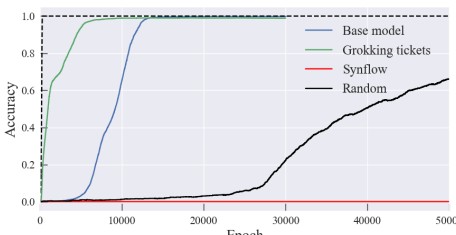

Figure 14: Comparing test accuracy of the different pruning methods

in subsection A.2). This implies that there are better pruning methods to find grokking tickets than magnitude pruning (Frankle & Carbin, 2019).

### D.5 CONTROLLING NETWORK SPARSITY

Similar to subsection 5.2, we compared the grokking tickets with the subnetworks identified differently using a transformer. Specifically, we tested three well-known pruning at initialization methods (Synflow (Tanaka et al., 2020)) and random pruning as baseline methods (see the detail in Appendix B). Figure 14 compares test accuracy of different pruning methods and grokking tickets. As a result of MLP experiments, all different methods perform worse than the base model. These results show that poor selection of the subnetworks hurts the generalization, suggesting that grokking tickets hold good properties beyond just a sparsity. A used similar parameters, setting the number of heads to four, making each head 32-dimensional

## E  VERIFICATION ACROSS DIVERSE ARCHITECTURES

In addition to the MLP and single-head transformer discussed in section 4,Appendix D, we also conducted verification with more multi-head transformers and deeper MLP to comprehensively validate our claims. Multi-head transformers used similar parameters exolained in subsection A.2, setting the number of heads to 4, making each head 32-dimensional. In deeper MLP experiments, we used a model one layer deeper than the one described in subsection 3.3. Figure 15 shows that comparison of base model (blue) and grokking tickets (green) on multi-head Transformer and deeper MLP. Both, like subsection 4.1, show accelerated generalization due to the discovery of Grokking tickets.

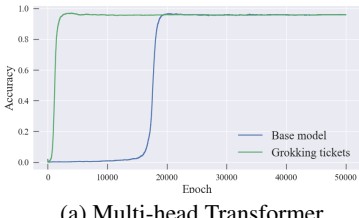 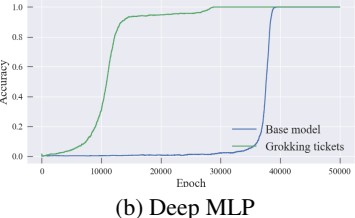

(a) Multi-head Transformer          (b) Deep MLP

Figure 15: Comparison of base model (blue) and grokking tickets (green) on multihead Transformer and deep MLP

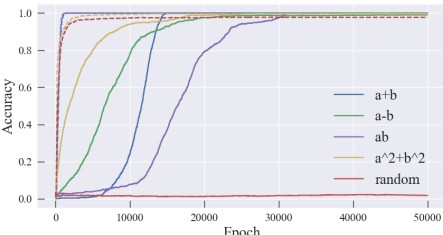

Figure 16: Comparison of base model (solid line) and grokking tickets (dashed line) on various tasks.

## F    VERIFICATION ACROSS DIVERSE TASKS

Following (Power et al., 2022), we tried several other operations (for a prime number $p = 67$).

$$a + b \quad (mod\ p) \quad for\ 0 < a, b < p$$
$$a - b \quad (mod\ p) \quad for\ 0 < b < a < p$$
$$ab \quad (mod\ p) \quad for\ 0 < a, b < p$$
$$a^2 + b^2 \quad (mod\ p) \quad for\ 0 < a, b < p$$

Figure 16 shows comparison of base model (solid line) and grokking tickets (dashed line) with various operations. Similar to section 4, it is evident that Grokking tickets reach generalization faster than the base model on various task , indicating that the analysis is independent of the task. We also tested random label and of course, the accuracy is random.

## G    GROKKING TICKETS FIND GOOD REPRESENTATION

Liu et al. (2022; 2023); Nanda et al. (2023) describe that representation learning is key to grokking. In particular, within the context of modular addition tasks, the topology of the embeddings tends to be circles or cylinders (see detail in subsection 6.2). In this section, we analyze the relationship between grokking tickets and representation learning, both qualitatively and quantitatively.

### G.1    PERIODIC PATTERNS

In the modular addition task, it is known that for certain neurons in the network, both the input-side weights and the output-side weights exhibit periodicity (Nanda et al., 2023; Pearce et al., 2023). Following Pearce et al. (2023), we investigated the periodicity of each neuron by plotting the neuron direction of the weight matrix ($W_{in}$) obtained by multiplying $W_{emb}$ and $W_{inproj}$ (defined in 3.2.1). This two-dimensional weight matrix represents the input dimensions(67) on one axis and the

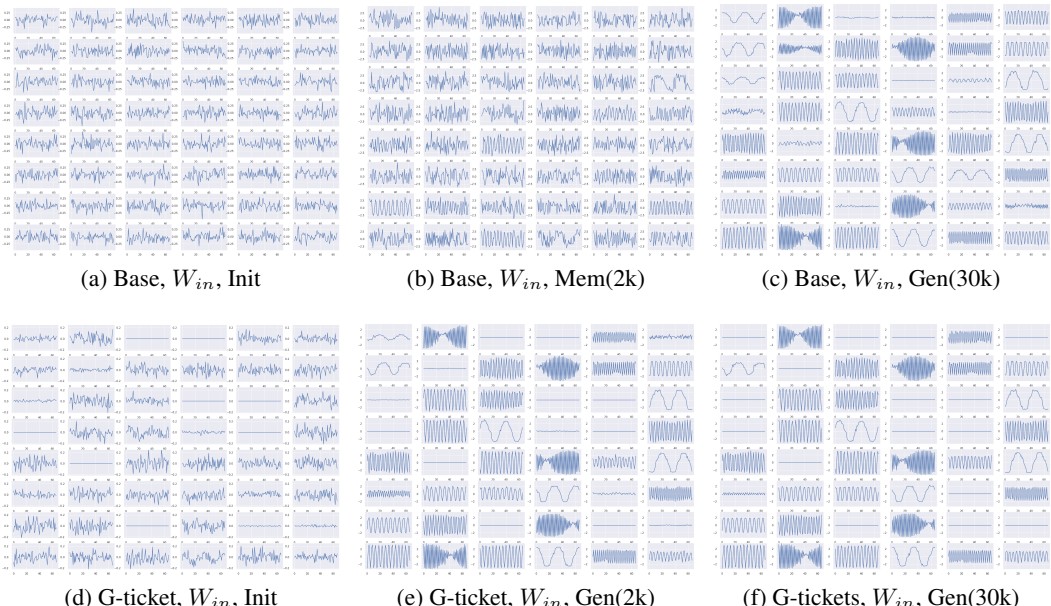

Figure 17: Comparison of base model and grokking tickets. The top/bottom row shows the transition of the input-side weights for each neurons of base model/grokking tickets. The horizontal axis represents input-dimention(67), while the vertical axis represents weight values.

number of neurons(48) on the other. Similarly, the output-side weight matrix ($W_{out}$) is obtained by multiplying $W_{unemb}$ and $W_{outproj}$. In this experiment, we used the same model as subsection 4.1.

Figure 17 shows the transition of the input-side weights for each neuron. While in the initialization and memorization phase, no specific frequency is observed in each neuron, after grokking(generalization phase), each neuron's weight of the base model shows a specific frequency. Additionally, some neurons receive almost no input (with weights approaching zero). We further analyzed these neurons in Appendix J. On the other hand, a grokking ticket has neurons with periodic weights similar to that observed after grokking appears at an early stage (2k epochs). The same is demonstrated for the output-side weights (Figure 18).

**Frequency characteristics of the neurons.** To examine the frequency characteristics of each neuron, we performed frequency decomposition of weight values for each neuron using a discrete Fourier transform. The discrete Fourier transform of $f(\boldsymbol{x})$ is as follows:

$$F(\boldsymbol{\omega}) = \sum_{x=0}^{N-1} f(x) \exp(-i\frac{2\pi\omega x}{N})$$

The number of sample points ($N$) is 67, the same as the input dimension. We conduct this discrete Fourier transform for both the input-side weights and the output-side weights of each neuron. Figure 19 shows that Fourier transform ($F(\boldsymbol{\omega})$ of the inside-weights ($W_{in}$) of each neurons. After grokking(generalization phase), the frequency characteristics of most neurons in the base model are prominent for a specific frequency. In grokking tickets, neurons responsive to specific frequencies emerge at an early stage. (2k epochs). The same applies to the output-side weights ($W_{out}$).

**Frequency entropy.** To quantitatively evaluate the representation learning of the base model and grokking tickets, we compared the frequency entropy of the weights for each neuron. The frequency entropy is as follows:

$$H(p) = -\frac{1}{K} \sum_{i=0}^{K} \sum_{t=0}^{\frac{N}{2}} p_i(t) \log p_i(t)$$

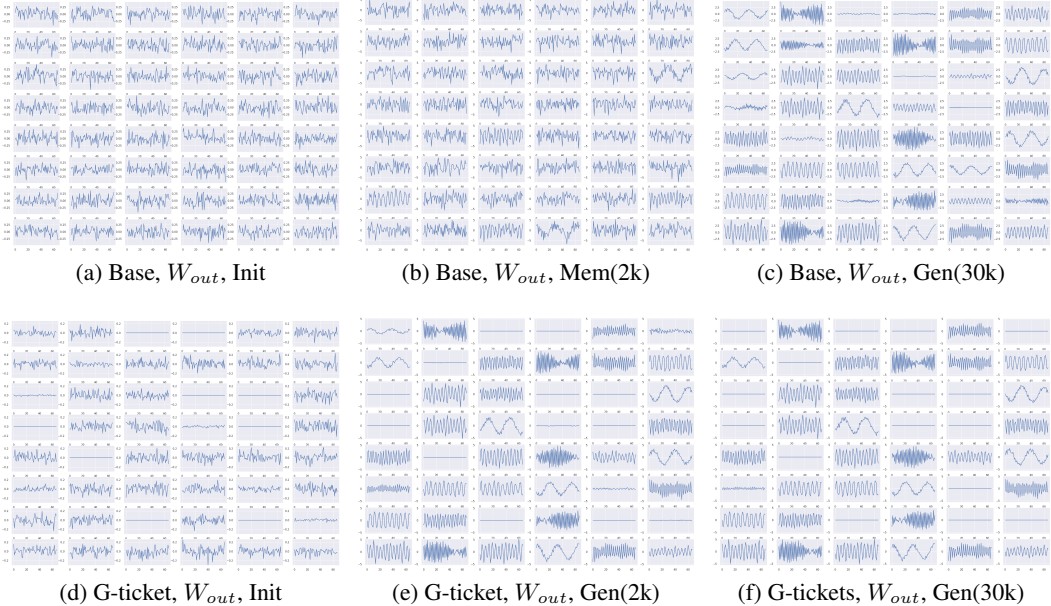

(a) Base, $W_{out}$, Init     (b) Base, $W_{out}$, Mem(2k)     (c) Base, $W_{out}$, Gen(30k)

(d) G-ticket, $W_{out}$, Init     (e) G-ticket, $W_{out}$, Gen(2k)     (f) G-tickets, $W_{out}$, Gen(30k)

Figure 18: Comparison of base model and grokking tickets. The top/bottom row shows the transition of the input-side weights for each neurons of base model/grokking tickets. The horizontal axis represents input-dimention(67), while the vertical axis represents weight values.

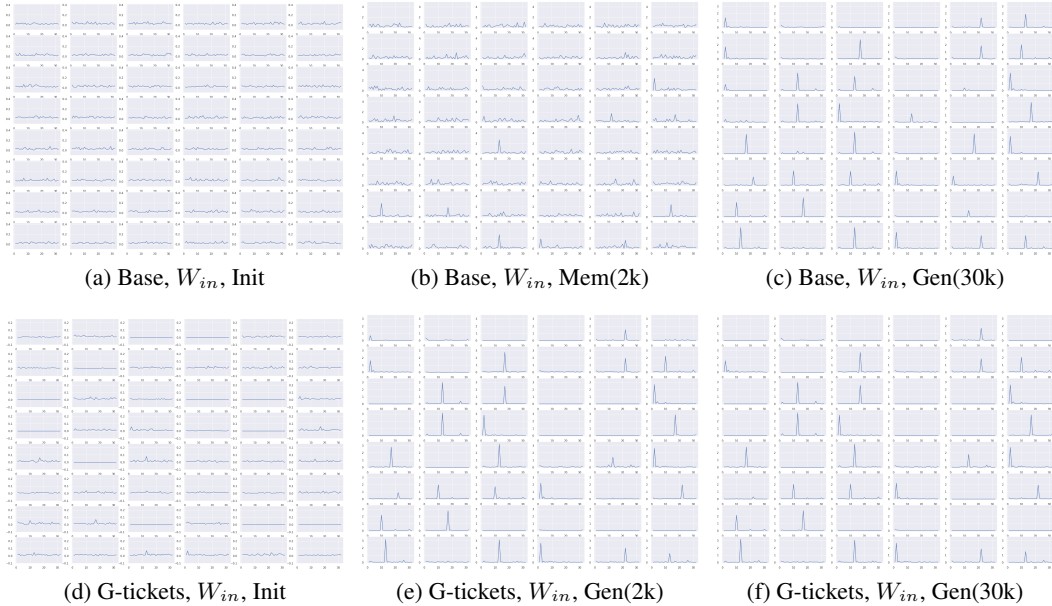

(a) Base, $W_{in}$, Init     (b) Base, $W_{in}$, Mem(2k)     (c) Base, $W_{in}$, Gen(30k)

(d) G-tickets, $W_{in}$, Init     (e) G-tickets, $W_{in}$, Gen(2k)     (f) G-tickets, $W_{in}$, Gen(30k)

Figure 19: Comparison of base model and grokking tickets. The top/bottom row shows the transition of the frequency decomposition of weight values for each neurons of base model/grokking tickets. The horizontal axis represents frequency, while the vertical axis represents amplification.

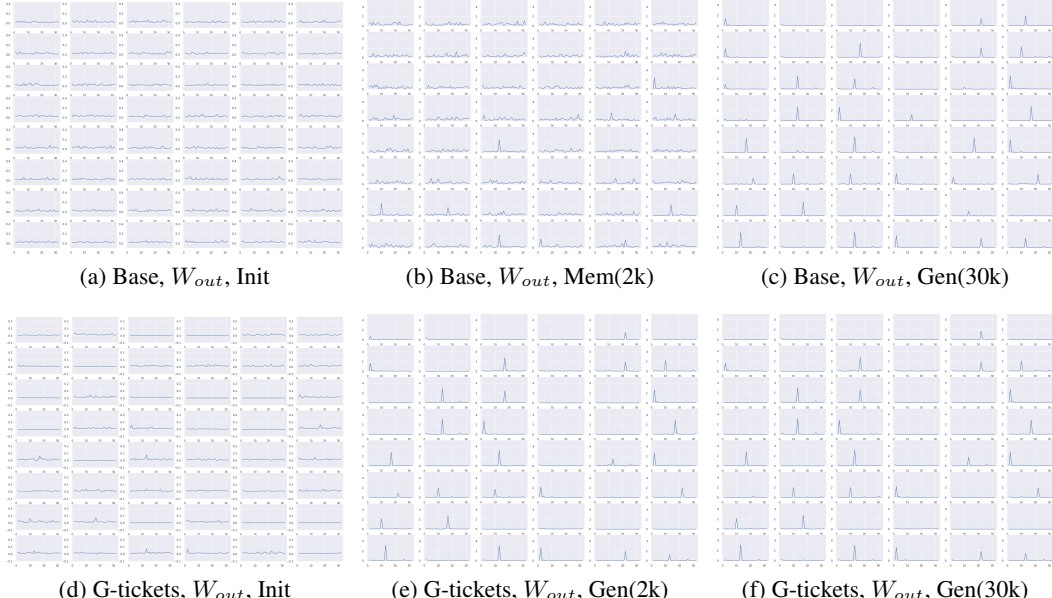

(a) Base, $W_{out}$, Init      (b) Base, $W_{out}$, Mem(2k)      (c) Base, $W_{out}$, Gen(30k)

(d) G-tickets, $W_{out}$, Init      (e) G-tickets, $W_{out}$, Gen(2k)      (f) G-tickets, $W_{out}$, Gen(30k)

Figure 20: Comparison of base model and grokking tickets. The top/bottom row shows the transition of the frequency decomposition of weight values for each neuron of base model/grokking tickets. The horizontal axis represents frequency, while the vertical axis represents amplification.

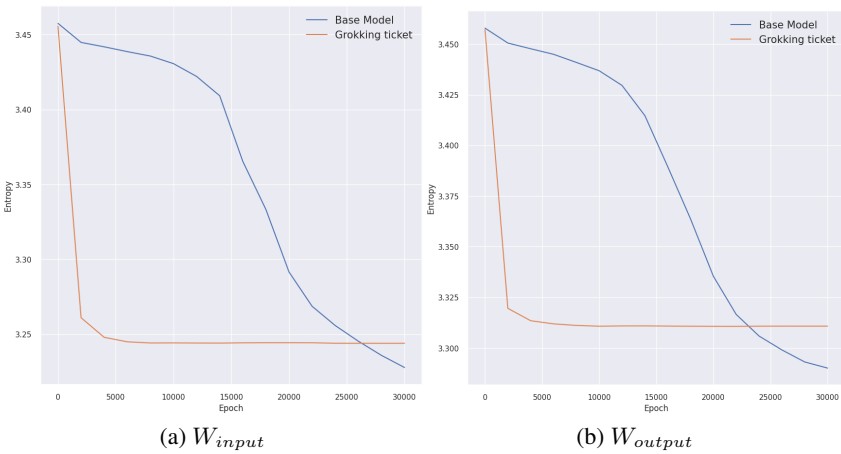

(a) $W_{input}$      (b) $W_{output}$

Figure 21: The entropy of base model and grokking ticket2.

The The number of neurons ($K$) is 48, the number of frequency points ($\frac{N}{2}$) is 33 and $p_i(t)$ is the normalized value at the $t^{th}$ sample point for the $i^{th}$ neuron. In Figure 21, the left panel show the transition of the base model and grokking tickets. Grokking tickets exhibit a rapid decrease in entropy compared to the base model. This indicates that the complexity of the frequency characteristics decreases in grokking tickets, leading to the acquisition of a simpler representation. The same is demonstrated for the output-side weights.

**Why is grokking tickets able to acquire a good representation?** One of the hypotheses related to this result is *The Strong Lottery Ticket Hypothesis* (SLTH) (Ramanujan et al., 2020). SLTH claims the existence of a subnetwork in a sufficiently large, randomly initialized neural network that approximates some target neural networks without the need for training. The results indicate that masking the initial values at a certain pruning rate promotes the acquisition of better representations. This

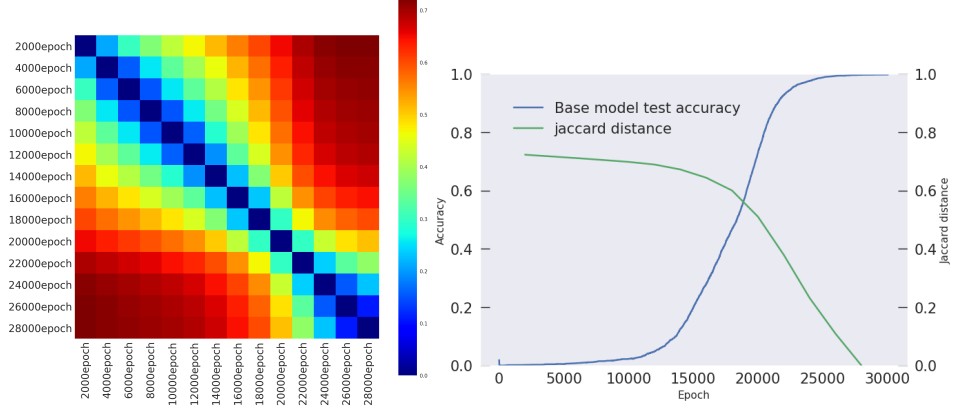

(a) Jaccard distance between epochs     (b) The relationship test accuracy and Jaccard distance

Figure 22: The distance between epochs at which Grokking tickets were acquired (left). The changes in Jaccard distance and accuracy between the Grokking tickets acquired after generalization and each epoch (right).

implies, similar to SLTH, that the random initial values contain certain reasonably good representations. (that certain combinations of initial values can represent reasonably good representations.) This also implies that there exists a generalizable solution in the vicinity of the initial values obtained by grokking tickets. In other words, the operation termed "grokking tickets" moves the initial values to the vicinity of a generalizable solution.

## H  STRUCTURAL SIMILARITY

Our paper solely contains empirical analyses, lacking theoretical analysis or profound analysis. Especially, there has been no analysis of how Grokking tickets are acquired during the process of grokking; comparisons have been made solely based on accuracy. Following (Paganini & Forde, 2020), we used Jaccard distance (Jaccard, 1901) to measure the similarity of structures. We measured the distance between the masks of two masked networks $(\boldsymbol{m}_1, \boldsymbol{m}_2)$ using the following metric:

$$d_j(\boldsymbol{m}_1, \boldsymbol{m}_2) = 1 - \frac{|\boldsymbol{m}_1 \cap \boldsymbol{m}_2|}{|\boldsymbol{m}_1 \cup \boldsymbol{m}_2|}$$

As in subsection 4.2, we measured the Jaccard distance of the masks when varying the epoch at which Grokking tickets are acquired in Figure 22. When epochs are close together, the distance between masks is small (indicating high similarity), showing that the structure is gradually acquired. Similar to the results in Section subsection 4.2, it is evident that Grokking tickets are discovered gradually, and the generalization performance also improves gradually.

## I  RE-DENSE NETWORK EXPERIMENTS

To respond to a reviewer's concern that the lottery tickets learn faster than the original dense model has already been demonstrated in the original LTH paper and many other previous works. Following (He et al., 2022; Han et al., 2017), we conducted experiments using Grokking tickets, where we applied the mask only to the initial values and not during the course of learning. We refer to this model as the 'Re-Dense model'. By doing this, the number of parameters optimized by the optimizer becomes equal in both the Dense and Re-Dense models, differing only in their initial values. Figure 23 shows that the test accuracy of Base model, Grokking tickets, and Re-Dense Model. Base Model and Re-Dense Model have the same number of parameters, with the only difference being in their initial values (further, the difference is just the application of a mask to the same initial values). It is evident that they generalize faster than the Base Model. This indicates that by simply altering the combination of initial values (through applying a mask), we can move the initial values closer to the vicinity of a generalization solution. Furthermore, grokking is a phenomenon that occurs as it searches for such combinations.

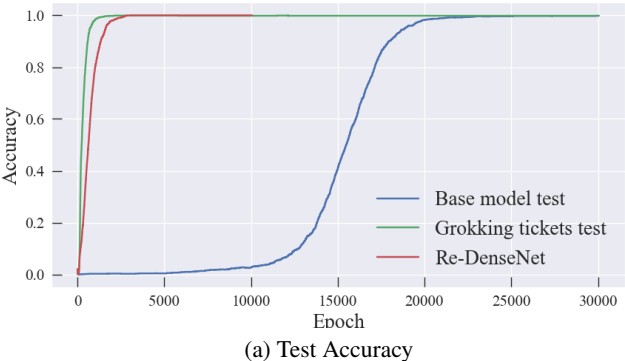

(a) Test Accuracy

Figure 23: Comparison of the test accuracy of Base model, Grokking tickets, and Re-Dense Model.

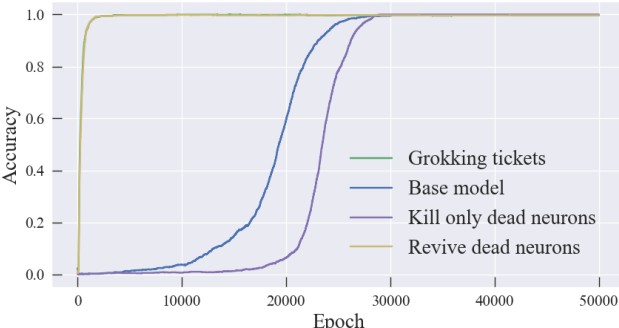

Figure 24: Test accuracy of Kill only the dead neurons and Revive the dead neurons.

## J    THE EFFECT OF DEAD NEURONS

In Appendix G, we investigated the periodicity of weights during grokking and the relationship with grokking tickets. During this process, neurons appeared whose input-side weights became all zeros. We refer to these neurons as "Dead neurons". To investigate the impact of these neurons on grokking tickets, we conducted the following two experiments:

1  Only the weights attached to the dead neurons were pruned. (Kill only the dead neurons)

2  The masks on the weights attached to the dead neurons were set to 1. (Revive the dead neurons)

In this experiment, the edges connected to dead neurons constituted 18.75% of the total. In Figure 24, when only the dead neurons were killed, grokking did not accelerate (Kill only dead neurons). This result implies that the acceleration of generalization is not simply due to a reduction in the parameters to be optimized, as would be the case if considering the impact of dead neurons. When the masks of dead neurons were set to 1 while keeping the other masks intact, the speed was almost the same as that of grokking tickets in Figure 24 (Revive dead neurons). This suggests that it's not the elimination of neurons that matters, but rather the significance lies in the other structures. It indicates that a kind of functional bias towards the task is being acquired as a structure.