# OpenReview forum: "Grokking Tickets: Lottery Tickets Accelerate Grokking"
_ICLR.cc/2024/Conference — Submitted to ICLR 2024_

### Official Review · Reviewer_tMC9 · 2023-10-16

**Soundness:** 2 fair
**Presentation:** 3 good
**Contribution:** 2 fair
**Rating:** 3
**Confidence:** 4

**Summary:**

The paper have conducted a set of experiments that demonstrate the effectiveness of grokking tickets in accelerating the generalization process. The results of the study suggest that the transition phase between memorized and generalized solutions corresponds to the exploration of good subnetworks, or lottery tickets.

**Strengths:**

1. The paper delves into the connection between grokking and LTH, demonstrating the grokking tickets hold good properties beyond just a sparsity.

2. Experimental results validate their findings.

**Weaknesses:**

1. The writing needs improving, and at times, I find it hard to follow the text. For instance, the term "grokking tickets" is used 12 times before its formal definition presented, which presents a certain impediment to readers.

2. I am uncertain about the practical significance of the findings. Specifically, I understand that one of the conclusions is that tickets found during the $C_{mem}$ phase exhibit better performance. However, this seems to align with the common practice in LTH research~\cite{chen2021unified, gan2022playing}, where networks are thoroughly trained and then pruned based on optimal validation scores during each pruning iteration. Therefore, the question arises as to what contribution the findings in the paper make to the discovery of improved lottery tickets.

3. The experimental results in Section 5.2 appear to support the author's claims that poor selection of subnetworks harms generalization. However, as mentioned in 2, the grokking tickets results seem to represent the outcomes of LTH approaches. It appears that the author has primarily validated that LTH with IMP can outperform most PaI methods.

4. While the existing results do indeed support the author's assertions, the use of overly simplistic tasks (modular addition and MNIST classification), basic baselines (MLP and single-head Transformer), and a lack of additional experimental configurations (such as performance under multi-layer MLPs or multi-head attention) leave me with reservations regarding the validity of the conclusions.

**Questions:**

1. The paper could be better organized. To name a few, in 2.1 ‘abuse’ -> ‘abuse of’; in 5.1 ‘leaders’->’readers’. Excessive space is devoted to some trivial aspects, such as formulas for weight initialization.

2. It is advisable to include additional backbones and experimental setups to comprehensively validate the claims.

---

> ### Author Response · Authors · 2023-11-19
> **Response to Reviewer tMC9**
>
> Thank you for your insightful comments. We will answer your questions to address your concerns.
>
> ## Writing
> >The writing needs improving, and at times, I find it hard to follow the text. For instance, the term "grokking tickets" is used 12 times before its formal definition presented, which presents a certain impediment to readers.
> >The paper could be better organized. To name a few, in 2.1 ‘abuse’ -> ‘abuse of’; in 5.1 ‘leaders’->’readers’. Excessive space is devoted to some trivial aspects, such as formulas for weight initialization.
>
> As you pointed out, we have defined Grokking tickets in the Abstract and at the beginning of the Introduction. Additionally, for readability, we have made changes to the writing, such as defining comparison models and correcting typographical errors.
>
> ## Contribution to Lottery Tickets (LT).
> >Therefore, the question arises as to what contribution the findings in the paper make to the discovery of improved lottery tickets.
>
> The novelty of this research in relation to the Lottery Tickets hypothesis is as follows: (1) Lottery Tickets as an explanation for the Grokking phenomenon. (2) Lottery Tickets that acquire good structures for generalization, not just a reduction in parameters.
> - Regarding point (1), in Appendices G and H, we conducted analyses on how Grokking tickets are discovered.
> In Appendix G, building upon prior research [Nanda 2022] [Zhong 2023], we investigated the structures that emerge during the transition phase in the modular addition task. In the task of modular addition, it is known that periodic representations are necessary for generalization. Our paper also demonstrates similar results, and it was found that in the case of Grokking tickets, good representations are acquired more quickly.
> In Appendix H, following the [Paganini 2020], we examined the structures acquired mid-way through grokking using the Jaccard distance [P. Jaccard 1901] . As can be seen from Section 4, the results show that structures which gradually generalize (lottery tickets) are being acquired. Especially during the transition phase, the distance of the masks decreases rapidly, indicating that the discovery of structures is important for generalization.
>
> - Regarding point (2),  in Appendices I and J, We conducted an analysis demonstrating that generalization is accelerated not merely due to the reduction of parameters through Lottery Tickets, but through the discovery of good structures.
> In Appendix I, to respond to the reviewer's concern that the lottery tickets learn faster than the original dense model has already been demonstrated in the original LTH paper and many other previous work. We conducted experiments using Grokking tickets, where we
> applied the mask only to the initial values and not during the course of learning. We refer to this model as the ’Re-Dense model’. By doing this, the number of parameters optimized by the optimizer becomes equal in both the Dense and Re-Dense models, differing only in their initial values.  Base Model and Re-Dense Model have the same number of parameters, with the only difference being in their initial values (further, the difference is just the application of a mask to the same initial values). It is evident that they generalize faster than the Base Model. In Appendix J, we conducted additional experiments. One experiment involved masking only the completely 'dead' units (18%) of Grokking tickets. If Grokking tickets were accelerating generalization simply by reducing the number of learnable parameters, as suggested by existing studies, then this method should also accelerate grokking. However, as shown in Appendix J, this method not only did not accelerate generalization but actually slowed it down. Conversely, it was also confirmed that reviving the completely dead units in Grokking tickets does not change the speed of generalization compared to the original Grokking tickets.
>
> We believe these results provide important analysis for research in the Lottery Ticket (LT) field.
>
> ## The Significance of the PaI Experiment.
> >the grokking tickets results seem to represent the outcomes of LTH approaches. It appears that the author has primarily validated that LTH with IMP can outperform most PaI methods.
>
> As you mentioned, it is known that IMP is superior to PaI. However, we did not intend to claim that IMP is superior to PaI, rather, we employed other pruning methods to demonstrate that Grokking tickets are beneficial not just in terms of sparsity, but also as a structure (refer to section 5.2).

---

> > ### Author Response · Authors · 2023-11-19
> > **Response2 to Reviewer tMC9**
> >
> > ## Additional Experiments
> > > While the existing results do indeed support the author's assertions, the use of overly simplistic tasks (modular addition and MNIST classification), basic baselines (MLP and single-head Transformer), and a lack of additional experimental configurations (such as performance under multi-layer MLPs or multi-head attention) leave me with reservations regarding the validity of the conclusions.
> > >It is advisable to include additional backbones and experimental setups to comprehensively validate the claims.
> >
> > Following the reviewer's comments, we have added experiments with wide MLPs and multi-head Transformers in Appendix F.
> > Consistent with the claims in the main text, it becomes clear in the case of Grokking tickets that the speed of generalization is accelerated.
> >
> > ## Reference
> > - Paganini, Michela, and Jessica Zosa Forde. "Bespoke vs. Pr\^ et-\a-Porter Lottery Tickets: Exploiting Mask Similarity for Trainable Sub-Network Finding." arXiv preprint arXiv:2007.04091 (2020).
> > - P. Jaccard. Etude de la distribution florale dans une portion des alpes et du jura. Bulletin de la Societe Vaudoise des Sciences Naturelles, 37:547–579, 01 1901. doi:10.5169/seals-266450.

---

> > > ### Author Response · Authors · 2023-11-22
> > > **A Reminder to reveiwer tMC9**
> > >
> > > Thank you again for your constructive comments.
> > > The discussion period will end soon, so please let us know if you have further comments about our reply.

---

### Official Review · Reviewer_DeHf · 2023-10-26

**Soundness:** 2 fair
**Presentation:** 2 fair
**Contribution:** 2 fair
**Rating:** 3
**Confidence:** 4

**Summary:**

The paper analyzes a phenomenon called ``Grokking'' by using the ``Lottery Ticket Hypothesis'' as a tool to uncover the reasons for Grokking. Grokking here refers to delayed generalization, i.e., test accuracy grows to its peak value long after the network has fit the training. The hypothesis examined here is that neural network training results in a sparse network. Sparsity may then be used to explain the reason behind delayed generalization. The paper conducts an empirical analysis with modular addition using a small Transformer  and briefly studies an MLP with large initialization as suggested by Liu et. al.  Ablations are conducted that show that the grokked solution checkpoint generalizes faster than checkpoints collected at earlier points during optimization.

**Strengths:**

- The paper uses the lens of sparsity to study Grokking. This is a promising avenue of exploration
- The empirical analysis for modular addition asks and then attempts to answer very sensible and relevant questions.
  - The choice of checkpoints suggest that useful sparsity occurs later on in optimization (close to or after solution is grokked)
  - Ablations with fixed norm and fixed sparsity are useful to readers interested in Grokking
  - The choice of pruning appears to make a difference. Checkpoints after generalization perform better than other algorithms used in pruning literature

**Weaknesses:**

- The paper attempts to make a connection to the Lottery Ticket Hypothesi (LTH). Sparsity is a reasonable hypothesis that has been explored by Merrill et. al. previously in literature. However, I am not convinced that connecting LTH to this work is necessary. Merrill 2023 make observations about sparse networks without invoking LTH
- Nanda et. al. (Nanda 2023) show that the network after generalization consists of a few sinusoids, i.e., finds a sparse solution. So sparsity being an explanation has been shown in prior literature. Also suggest that the authors consult Gromov 2023 for the solution found a

- The paper notes in Section 5.1( ablation of weight norms) that a subnetwork has increased weight norm after a generalizing solution is found by gradient descent while other weights end up with smaller norms. This is the same observation made in Merrill 2023 where the authors study subset parity learning problem
- The observation that the sparse generalized solution optimizes faster than the dense network while interesting is not critical to understanding Grokking. It is known that regularization or lowering capacity of models via regularization does help shorten the time between fitting and generalization in algorithmic datasets. This observation falls in-line with the above
- Given the empirical nature of the paper, the number of datasets considered in the analysis appear to be inadequate. Power 2022 construct various algorithmic datasets. Gromov 2023 use a MSE solution with MLP for addition. Have the observations been confirmed in more settings than the ones considered in the paper?

- [Gromov 2023]  Grokking modular arithmetic
- [Meriill 2023] William Merrill, Nikolaos Tsilivis, and Aman Shukla. A tale of two circuits: Grokking as competi- tion of sparse and dense subnetworks, 2023.
- [Nanda 2023] Neel Nanda, Lawrence Chan, Tom Lieberum, Jess Smith, and Jacob Steinhardt. Progress measures for grokking via mechanistic interpretability, 2023.
- [Power 2022] Alethea Power, Yuri Burda, Harri Edwards, Igor Babuschkin, and Vedant Misra. Grokking: Gener- alization beyond overfitting on small algorithmic datasets, 2022.

**Questions:**

**Major Issues**

- Please see weakness section
- Using sparsity as an explanation for Grokking has been done before in other papers. How is the analyses presented in the paper adding to existing literature? Without a clear answer, the novelty of the work in the paper is insufficient for me to vote for acceptance at the conference.

**Minor Issues**

- Introduction
  -  Power et. al, observed Grokking in many algorithmic datasets including modular addition.
  -  Barak et. al. [2] demonstrated Grokking on subset parity dataset before Merrill et al.
  -  ``accelerate'' the grokking process may be confusing to readers outside the area of Grokking. Perhaps emphasize that the time to generalization is shorter?

- Page 7
  - leaders should be readers

[2] https://openreview.net/forum?id=8XWP2ewX-im

---

> ### Author Response · Authors · 2023-11-19
> **Response to Reviewer DeHf**
>
> Thank you for your insightful comments. We will answer your questions to address your concerns.
>
> ## The Reason for Using Lottery Tickets (LT).
> >I am not convinced that connecting LTH to this work is necessary. Merrill 2023 make observations about sparse networks without invoking LTH.
>
> To clarify the differences from prior research, we added a section on related works.
> As mentioned in section 6, while Merrill et. al  prunes neurons with smaller weight norms, we use the lottery ticket (LT) method for pruning weights. Besides, we tested arithmetic tasks where the importance of sparsity is not obvious and the standard benchmark in the grokking, while they have only conducted experiments in tasks like (n-k)-parity where the importance of sparsity is obvious.
> To be applicable to more general tasks and architectures, we need to use the Lottery Tickets for analysis of grokking using the Lottery Ticket (LT) method.
>
> ## Relation to Previous Research on Sparsity.
> >So sparsity being an explanation has been shown in prior literature.
>
> The novelty of our study compared to previous research can be summarized in four points: 1) Methodology (Lottery Tickets), 2) Tasks, 3) Architecture, and 4) Claims.
> 1) We conducted analysis using the Lottery Tickets (LT) method. The necessity of analysis through LT is as described above.
>
> 2) Our tasks are not (n-k) parity where the optimal structure being sparse  is self-evident, but we have also validated in more general tasks such as MNIST.
>
> 3) Unlike Merrill 2023 or Nanda 2022, we have not confined our validation to just one architecture; instead, we have tested across multiple architectures.
>
> 4) Furthermore, we assert that it's not just sparsity that's important, but the acquisition of good structures is crucial (refer to section 5.2, Appendix G and H).
>
> ## Differences from Merrill 2023.
> >This is the same observation made in Merrill 2023 where the authors study subset parity learning problem.
>
> Indeed, similar observations have been confirmed, but as mentioned above, our results indicate that not just sparsity, but also the acquisition of good structures is important for generalization.  In section 5.2 in the original version of our paper, we compared various methods fixed at the same level of sparsity, and these results showed that Grokking tickets have effects beyond just reducing parameters. This shows that in explaining grokking, it's not just sparsity that's key, but also the acquisition of good structures is crucial for generalization.
>
> ## The Novelty of Grokking Tickets.
> >It is known that regularization or lowering capacity of models via regularization does help shorten the time between fitting and generalization in algorithmic datasets.
>
> As suggested in existing research, one reason why generalization is faster with Grokking tickets is due to the reduction in the number of learnable parameters. However, our study suggests that grokking is not just about reducing the effective number of parameters being learned, but also about acquiring structures appropriate to the task. For instance, in section 5.2 in the original version of our paper, we compared various methods fixed at the same level of sparsity, and these results showed that Grokking tickets have effects beyond just reducing parameters.
> To analyze this in detail, we conducted additional experiments. One experiment involved masking only the completely 'dead' units (18%) of Grokking tickets (see Appendix J). If Grokking tickets were accelerating generalization simply by reducing the number of learnable parameters, as suggested by existing studies, then this method should also accelerate grokking. However, as shown in Appendix J, this method not only did not accelerate generalization but actually slowed it down. Conversely, it was also confirmed that reviving the completely dead units in Grokking tickets does not change the speed of generalization compared to the original Grokking tickets.
> We have added these discussions in the revised section 6.4.

---

> > ### Author Response · Authors · 2023-11-19
> > **Response2 to Reviewer DeHf**
> >
> > ## Additional Experiment
> > >  Have the observations been confirmed in more settings than the ones considered in the paper?
> >
> > Following the reviewers’ comments, we conducted experiments across various task configurations and with different architectures.
> > In Appendix D, similar to MLP experiments, we conduct experiments on single-head transformer architecture. Similar to the experiments conducted in the paper, similar results were obtained on single-head transformer architecture as well.
> > In Appendix E,  we conduct experiments on multi-head transformer architecture and multi-layer MLP. Similar to the experiments conducted in the paper, generalization is faster due to Grokking tickets. on multi-head transformer architecture and multi-layer MLP as well.
> > In Appendix F, referencing [Power 2022], we conduct experiments on diverse tasks. Similar to the experiments conducted in the paper, generalization is faster due to Grokking tickets　on diverse tasks.
> >
> >
> > ## The novelty
> > > Using sparsity as an explanation for Grokking has been done before in other papers. How is the analyses presented in the paper adding to existing literature?
> >
> > To reiterate, the novelty of our study compared to previous research(Merrill and Nanda) can be summarized in four points: 1) Methodology (Lottery Tickets), 2) Tasks, 3) Architecture, and 4) Claims.
> >
> > ## Reference
> > - Liu, Ziming, Eric J. Michaud, and Max Tegmark. "Omnigrok: Grokking beyond algorithmic data." arXiv preprint arXiv:2210.01117 (2022).
> > - Merrill, William, Nikolaos Tsilivis, and Aman Shukla. "A Tale of Two Circuits: Grokking as Competition of Sparse and Dense Subnetworks." arXiv preprint arXiv:2303.11873 (2023).
> > - Power, Alethea, et al. "Grokking: Generalization beyond overfitting on small algorithmic datasets." arXiv preprint arXiv:2201.02177 (2022).

---

> > > ### Comment · Reviewer_DeHf · 2023-11-22
> > > **Rebuttal response**
> > >
> > > I thank the authors for their rebuttal. My understanding now is that one of the central claims is that the Lottery Ticket Hypothesis (LTH)suggests acquisition of certain specific structures helps with grokking, i.e., the grokking lottery ticket. My understanding, based on reading all of the responses by the authors, is this insight flows naturally from the LTH. This part still feels incomplete as the paper does not shed any light on what can be said about the structures acquired after a model reaches low test loss (i.e., groks). Nanda et al (2023) have shown that for modular addition the structure can be quantified via Fourier coefficients for Transformers. Gromov (2023) provides an analytical solution for an MLP that is also characterized by Fourier coefficients for modular addition. I do not see any new insights being applied to grokking setup by using the LTH that can help researchers in the subfield of grokking/mechanistic interpretability make progress.  Hence, I plan to keep my score unchanged while also being open to hearing from other reviewers in further discussions. If the authors feel like I have missed something critical here and/or have new insights then I'd like to hear from them.

---

> > > > ### Author Response · Authors · 2023-11-22
> > > > **Response3 to Reviewer DeHf**
> > > >
> > > > We thank the feedback provided in your response.
> > > > > This part still feels incomplete as the paper does not shed any light on what can be said about the structures acquired after a model reaches low test loss (i.e., groks).
> > > >
> > > > We have described  the structures acquired after a model reaches low test loss in detail in the Appendix G and H.
> > > > Our model demonstrates the periodicity as described in Nanda et al (2023), Gromov (2023). Regarding Grokking tickets, it is evident that this periodic structure is acquired in the early stages of learning.
> > > > While these results are indeed suggested by prior research as you have mentioned, our work bridges the gap between studies indicating that sparsity contributes to generalization (grokking) and those suggesting the role of representational learning in grokking, offering new insights like this from the perspective of LTH (Lottery Ticket Hypothesis).

---

### Official Review · Reviewer_KZMf · 2023-11-01

**Soundness:** 3 good
**Presentation:** 2 fair
**Contribution:** 2 fair
**Rating:** 3
**Confidence:** 4

**Summary:**

This paper explores the effect of lottery tickets in the context of Grokking and finds that the Grokking tickets obtained during the generalization phase can accelerate the speed of the model to reach the generalization phase. The found that the speedup only happens when subnetworks are identified at the generalization solution. Whereas, there is no speed up when we try to identify subnetworks at initialization, or at the memorization solution or the transition between memorization  and generalization. In general, this is an interesting combination of LTH and Grokking but with some mediocre observations.

**Strengths:**

1. This paper explores the role of LTH in Grokking and make a good combination of LTH and Grokking.

2. The finding that only subnetworks discovered during the generalization phase can speedup the generalization process is reasonable, aligning with previous findings of LTHs.

3. Figure 1 clearly demonstrates the main message delivered by this paper.

4. They also demonstrate that it is possible to induce grokking without weight decay when using the grokking tickets.

**Weaknesses:**

1. My major concern is that while the combination of LTH with Grokking is new to the community, the empirical findings shown in this paper is somehow mediocre. For instance, i. it is not surprising to see that the subnetworks obtained during the generalization phase is crucial for grokking speedup.  ii. The lottery tickets learn faster than the original dense model has already been demonstrated in the original LTH paper and many other previous works. iii. The definition of the Grokking Tickets has no essential difference than the original Lottery Tickets and can be covered by the original ones since the original LTH does train the dense model to the end.

2.  Perhaps, the authors can emphasize the contribution of the papers from the perspective of Grokking. Why the grokking tickets are important for Grokking?

3. Besides the speedup, does Grokking tickets bring any performance benefits over the dense one?

**Questions:**

Please refer to the above weaknesses.

---

> ### Author Response · Authors · 2023-11-19
> **Response to Reviewer KZMf**
>
> Thank you for your insightful comments. We will answer your questions to address your concerns.
> ## Profound Analysis
> >it is not surprising to see that the subnetworks obtained during the generalization phase is crucial for grokking speedup.
>
> As you mentioned, the acceleration of grokking in a subnetwork is not surprising.
> To address your concern, we have added the following two analysis.
> 1) As other reviewers have pointed out, the simplest explanation for faster generalization is that the number of learnable parameters decreases. However, our analysis suggests that performance improvement is not just due to a reduction in parameters. Firstly, not any structure works effectively (refer to section 5.2), and secondly, even if only completely 'dead' neurons in grokking tickets (18%) are removed, learning does not accelerate but rather decelerates (refer to Appendix J). These results imply that it's not just about reducing the number of parameters, but the construction of some good structures contributes to the acceleration of generalization.
> 2)
> As stated in general response (2), to analyze how Grokking tickets are acquired during the course of learning, we examined the periodicity of weights (refer to Appendix G) and the similarity of masks (refer to Appendix H).
> Additionally, the results of the newly added analysis of representations in grokking tickets indicate that better representations are acquired earlier with grokking tickets. While the relationship between grokking and representation learning has been widely discussed, our study connects the perspectives of representation learning and structure acquisition in grokking.
>
>
> ## Novelty with Respect to Lottery Tickets.
> >The lottery tickets learn faster than the original dense model has already been demonstrated in the original LTH paper and many other previous works.
>
> As you mentioned, it is known that the lottery tickets learn faster than the original dense model.
> As mentioned above, we have further made the following claims and conducted experiments:
> 1) It's not the reduction in the number of parameters, but the acquisition of good structures that is crucial for generalization (refer to section 5 and Appendix I,J).
> 2) How good structures are acquired during learning (Appendix G,H).
> We have added similar discussions in the main text section 6.4 as well.
>
>
> ## The difference between Grokking Tickets and the original Lottery Tickets
> >The definition of the Grokking Tickets has no essential difference from the original Lottery Tickets and can be covered by the original ones since the original LTH does train the dense model to the end.
>
> As stated in General Response (2), we are not proposing a new LT method; rather, we are employing the Lottery Ticket (LT) method as a means to analyze grokking. Therefore, Grokking tickets are encompassed within the definition of LT.
> However, as mentioned above, experiments in Appendix I and J, demonstrate that Grokking tickets are not merely sparse (i.e., having fewer learning parameters), but also acquire good structures for generalization.
>
> ## The Reasons Why Grokking Tickets are Important for Grokking.
> >Perhaps, the authors can emphasize the contribution of the papers from the perspective of Grokking. Why the grokking tickets are important for Grokking?
>
> As mentioned in the General Response as well, in section 6, we have detailed the comparison between previous studies on grokking and our research. Especially in existing studies, explanations often focus on weight norms and weight decay [Liu 2022, Power 2022]. In contrast, our research not only examines weight norms but also focuses on the network structure. We determined that grokking occurs due to the discovery of lottery tickets.
> Although Merrill et al. emphasize the importance of sparsity, they do not extend their claim to the acquisition of good structures.
> Therefore, in response to the question 'Why are Grokking tickets important for Grokking?', our answer lies in the acquisition of good structures.
>
> ## Benefits over the Dense model
> >Besides the speedup, does Grokking tickets bring any performance benefits over the dense one?
>
> Of course, as we are finding extremely sparse solutions, I believe it is highly relevant from the perspective of learning efficiency. For instance, the model we used in this study maintained its generalization performance even after pruning about 81% of its edges. (Conversely, this led to an earlier occurrence of grokking.)
> As also mentioned in the General Response, the relationship with representation learning is conceivable, and our findings are likely to provide important insights from the perspective of model interpretability.

---

> > ### Author Response · Authors · 2023-11-19
> > **Reference**
> >
> > ## Reference
> > - Liu, Ziming, Eric J. Michaud, and Max Tegmark. "Omnigrok: Grokking beyond algorithmic data." arXiv preprint arXiv:2210.01117 (2022).
> > - Merrill, William, Nikolaos Tsilivis, and Aman Shukla. "A Tale of Two Circuits: Grokking as Competition of Sparse and Dense Subnetworks." arXiv preprint arXiv:2303.11873 (2023).
> > - Power, Alethea, et al. "Grokking: Generalization beyond overfitting on small algorithmic datasets." arXiv preprint arXiv:2201.02177 (2022).

---

> ### Comment · Reviewer_KZMf · 2023-11-20
> **I am not convinced by authors' response.**
>
> I thank the authors for your response. However, I am not convinced by the response. I don't think the faster generalization has any correlation with the number of learnable parameters decrease. And the construction of some good structures contributes to the acceleration of generalization is not something new to LTH. It has very little merits to the concept of LTH.
>
> After seeing other reviewers' comment, I believe the merit of this work to Grokking is also limited. Therefore, I decide to decrease my score to reject.

---

### Official Review · Reviewer_qTo9 · 2023-11-07

**Soundness:** 3 good
**Presentation:** 3 good
**Contribution:** 3 good
**Rating:** 6
**Confidence:** 2

**Summary:**

The paper investigates the grokking phenomenon in neural networks through the lens of the lottery ticket hypothesis, positing that identifying optimal sparse subnetworks ("lottery tickets") is crucial for the transition from memorization to generalization. The authors present experiments using MLP and Transformer architectures on tasks like modular addition and MNIST classification to demonstrate that these identified subnetworks can significantly accelerate grokking.

**Strengths:**

**Novel Approach:** The paper introduces a novel concept of "grokking tickets" within the context of the lottery ticket hypothesis, which is an original contribution to the understanding of neural network learning dynamics.

**Experimental Evidence:** Initial experimental results indicate that the identified subnetworks do indeed accelerate the grokking process, which could have implications for the efficiency of training neural networks.

**Clarity of Presentation:** The paper is well-structured and presents its methodology and findings clearly, making a case for the importance of subnetwork identification in neural network training.

**Weaknesses:**

**Theoretical Underpinning:** The paper lacks a comprehensive theoretical framework that explains why and how grokking tickets work, leaving the reader to infer the underlying principles from empirical observations.

**Limited Experimental Scope:** The experiments are confined to a narrow set of tasks and architectures, which might not fully demonstrate the generalizability of the proposed method.

**Lack of In-Depth Analysis:** The paper does not provide an in-depth analysis of the scalability of the approach or a comparison with other state-of-the-art methods across varied settings.

**Comment on References:** While the paper presents a novel approach to understanding the grokking phenomenon in neural networks, the reference list does not appear to be fully comprehensive. It would strengthen the paper to include a broader range of sources that contextualize the work within the larger body of research on neural network pruning, generalization, and learning dynamics.

**Questions:**

1. Can the authors elaborate on the theoretical foundations that might explain the observed acceleration in learning due to grokking tickets?

2. Would the authors consider expanding their experimental evaluation to include a wider variety of tasks and network architectures to confirm the robustness of their findings?

3. How do the authors envision the scalability of the proposed method, and how does it compare with other pruning or training acceleration techniques in terms of computational efficiency and performance?

---

> ### Author Response · Authors · 2023-11-19
> **Response to Reviewer qTo9**
>
> Thank you for your insightful comments. We will answer your questions to address your concerns.
> ## Theoretical Underpinning
> >Theoretical Underpinning
> >Can the authors elaborate on the theoretical foundations that might explain the observed acceleration in learning due to grokking tickets?
>
> To address your concern, we have added the following two analisis.
> As other reviewers have pointed out, the simplest explanation for faster generalization is that the number of learnable parameters decreases. However, our analysis suggests that performance improvement is not just due to a reduction in parameters. Firstly, not any structure works effectively (refer to section 5.2), and secondly, even if only completely 'dead' neurons in grokking tickets (18%) are removed, learning does not accelerate but rather decelerates (refer to Appendix J). These results imply that it's not just about reducing the number of parameters, but the construction of some good structures contributes to the acceleration of generalization.
>
> ## Depth Analysis
> >Lack of In-Depth Analysis
>
> As stated in general response (2), to analyze how Grokking tickets are acquired during the course of learning, we examined the periodicity of weights (refer to Appendix G) and the similarity of masks (refer to Appendix H).
> Additionally, the results of the newly added analysis of representations in grokking tickets indicate that better representations are acquired earlier with grokking tickets. While the relationship between grokking and representation learning has been widely discussed, our study connects the perspectives of representation learning and structure acquisition in grokking.
>
> ## Expand Experimental Scope
> >Limited Experimental Scope
> >Would the authors consider expanding their experimental evaluation to include a wider variety of tasks and network architectures to confirm the robustness of their findings?
>
> Following the reviewers’ comments, we conducted experiments across various task configurations and with different architectures.
> In Appendix D, similar to MLP experiments, we conduct experiments on single-head transformer architecture. Similar to the experiments conducted in the paper, similar results were obtained on single-head transformer architecture as well.
> In Appendix E,  we conduct experiments on multi-head transformer architecture and multi-layer MLP. Similar to the experiments conducted in the paper, generalization is faster due to Grokking tickets. on multi-head transformer architecture and multi-layer MLP as well.
> In Appendix F, referencing [Power 2022], we conduct experiments on diverse tasks. Similar to the experiments conducted in the paper, generalization is faster due to Grokking tickets　on diverse tasks.
>
> ## The novelty of our study
> >​​Comment on References
>
> The novelty of our study compared to prior research was unclear. Therefore, in section 6, we summarized the superiority and novelty of our research over previous studies on grokking. Specifically, our main assertion delves deeper than the analysis of weight norms and weight decay in prior research [Liu 2022, Power 2022], pinpointing structural exploration as the underlying cause.
>
> ## The scalability of our  method
> >How do the authors envision the scalability of the proposed method, and how does it compare with other pruning or training acceleration techniques in terms of computational efficiency and performance?
>
> As mentioned in the General Response, in the Appendix G and H, unlike other pruning methods, we investigated whether the network structure changes similarly during grokking and generalizes, from the perspective of Lottery Tickets (LT). This analysis is beneficial not only for research on efficient learning methods but also for the interpretability of models.

---

> > ### Author Response · Authors · 2023-11-22
> > **A Reminder to reveiwer qTo9**
> >
> > Thank you again for your constructive comments. The discussion period will end soon, so please let us know if you have further comments about our reply.

---

### Official Review · Reviewer_NUTh · 2023-11-12

**Soundness:** 3 good
**Presentation:** 2 fair
**Contribution:** 3 good
**Rating:** 6
**Confidence:** 4

**Summary:**

This work provided a new perspective for analyzing the grokking phenomenon, that is the lottery ticket hypothesis.

The authors first showed an interesting observation they made. They trained a neural network to the grokking stage and then performed one-shot pruning. They called the subnetwork obtained as “grokking ticket”. The authors observed that the grokking ticket can significantly shorten the training epochs needed to achieve the grokking stage, compared to the dense network.

Furthermore, the authors conducted a series of ablation experiments to deconstruct the effects brought by the grokking tickets, including:
* Pruning stage: tickets obtained before the grokking stage won’t accelerate the occurrence of the grokking stage.
* Weight norms: as the grokking phenomenon has been connected to the weight norms in the literature, the authors also compared the grokking tickets with dense networks whose weights are scaled to have similar norms. The results showed no acceleration through weight scaling.
* Weight decay: the authors showed with numerical results that the grokking tickets at appropriate pruning ratios can waive the necessity of weight decay, which has been assumed necessary for grokking to happen.

**Strengths:**

+ This work provided a new brandnew perspective for investigating and understanding the grokking phenomenon, that is the sparse network structure. The authors made interesting observations on the acceleration of grokking brought by pruning, which connects the seemingly disjoint topics.

+ The authors provided results of well-designed experiments that decoupled grokking tickets and weight norms (thus weight decaying), which have been assumed to be the key of grokking, throwing light on a higher level of property that grokking could possess.

+ Moreover, this work also provided a perspective of using grokking phenomenon to understand pruning and specifically lottery tickets. Many previous studies have shown that SNIP, GraSP and SynFlow achieved similar accuracies on classification tasks but Fig. 7 in this work showed very different behaviors with these methods.

**Weaknesses:**

- While this work made a lot of interesting observations, profound analysis and investigation behind empirical results are lacking.

- The writing quality of this work is not satisfactory.

**Questions:**

n/a

---

> ### Author Response · Authors · 2023-11-19
> **Response to Reviewer NUTh**
>
> Thank you for your insightful comments. We will answer your questions to address your concerns.
>
> ## Profound Analysis
> >While this work made a lot of interesting observations, profound analysis and investigation behind empirical results are lacking.
> Following the reviewer comments, we conducted three additional analyses to investigate the implication behind empirical results.
>
> (a) Analyzing similarity of structure (lottery tickets) among memorization, transition, and generalization phase of training.
> Appendix H, following the [Paganini 2020], we examined the structures acquired mid-way through grokking using the Jaccard distance [P. Jaccard 1901] . As can be seen from Section 4, the results show that structures which gradually generalize (lottery tickets) are being acquired. Especially during the transition phase, the distance of the masks decreases rapidly, indicating that the discovery of structures is important for generalization.
>
> (b) Connection between grokking tickets to existing analysis.
> Recently,  [Nanda 2022] investigated the weights of networks during grokking, and found that the weights exhibit clear periodic patterns after the grokking occurs. We follow the similar experiments to connect the grokking tickets and the representation learning perspective of the grokking in Appendix G. The results show that the grokking tickets help to learn task-appropriate representations, i.e. unlike the dense model, grokking tickets help to learn the periodicity in weights at the early stages of training.
>
> ## writing quality
> >The writing quality of this work is not satisfactory.
>
> Thank you for carefully reading our paper. To improve the clarity, we proof-read the paper again and made necessary revisions. A significant change is the addition of a section on related work in Section 6. We also have clearly defined grokking tickets and other comparison models in the abstract and introduction.

---

> > ### Author Response · Authors · 2023-11-22
> > **A Reminder to reveiwer NUTh**
> >
> > Thank you again for your constructive comments. The discussion period will end soon, so please let us know if you have further comments about our reply.

---

### Author Response · Authors · 2023-11-19
**General response**

Thank you again to all reviewers for carefully reading our paper and giving insightful comments. Following the reviewers' comments, we conducted additional experiments and analyses, and revised the writing for improving clarity.
The figures are listed in the attached PDF.

## (1) Additional experiments: expanding tasks and architectures (reviewer qTo9, KZMf, DeHf, tMC9).
Following the reviewers comments, we conducted experiments across various task configurations (e.g., other arithmetic tasks) and with different architectures (including multihead transformer and deeper MLP).
In Appendix D, similar to MLP experiments, we conduct experiments on single-head transformer architecture. Similar to the experiments conducted in the paper, similar results were obtained on single-head transformer architecture as well.
In Appendix E,  we conduct experiments on multi-head transformer architecture and multi-layer MLP. Similar to the experiments conducted in the paper, generalization is faster due to Grokking tickets. on multi-head transformer architecture and multi-layer MLP as well.
In Appendix F, referencing [Power 2022], we conduct experiments on diverse tasks. Similar to the experiments conducted in the paper, grokking tickets shorten the time to generalization on diverse tasks.

## (2) Additional analysis: a) How Grokking ticket’s structures are acquired and b) Acceleration of generalization due to good structures.  (reviewer NUTh, qTo9, tMC9).
### a) As pointed out in the reviewer's comments, our paper solely contains empirical results, lacking profound analysis. To address this concern, we analyzed how Grokking tickets are acquired in Appendix G and H.
In Appendix G, building upon prior research [Nanda 2022] [Zhong 2023], we investigated the structures that emerge during the transition phase in the modular addition task. In the task of modular addition, it is known that periodic representations are necessary for generalization. Our paper also demonstrates similar results, and it was found that in the case of Grokking tickets, good representations are acquired more quickly.
In Appendix H, following the [Paganini 2020], we examined the structures acquired mid-way through grokking using the Jaccard distance [P. Jaccard 1901] . As can be seen from Section 4, the results show that structures which gradually generalize (lottery tickets) are being acquired. Especially during the transition phase, the distance of the masks decreases rapidly, indicating that the discovery of structures is important for generalization.
### b) To address the reviewer's concern that the lottery tickets learn faster than the original dense model has already been demonstrated in the original LTH paper and many other previous work, we conducted two experiments in Appendix I and J.
In Appendix I, we conducted experiments using Grokking tickets, where we
applied the mask only to the initial values and not during the course of learning. We refer to this model as the ’Re-Dense model’. By doing this, the number of parameters optimized by the optimizer becomes equal in both the Dense and Re-Dense models, differing only in their initial values.  Base Model and Re-Dense Model have the same number of parameters, with the only difference being in their initial values (further, the difference is just the application of a mask to the same initial values). We have demonstrated that Grokking tickets perform faster than the base model due to possessing good structures.
In Appendix J, we investigated dead neurons, specifically those in Grokking tickets where all input-side weights become zero. The results show that the impact of dead neurons on Grokking tickets is minimal. It indicates that Grokking tickets acquire a good structure for the task, meaning that the accelerated learning is not simply due to a reduction in the parameters to be optimized.
We believe that these analyses contribute not only to the understanding of the grokking phenomenon but also to the research on Lottery Tickets.

---

> ### Author Response · Authors · 2023-11-19
> **General response 2**
>
> ### (3) Clarifying the contribution to the grokking studies.
> Several reviewers commented that it is obvious that lottery tickets accelerate generalization. To be clear, our main focus is not saying the lottery ticket accelerates the generalization. Rather, our main goal is to investigate the inner-working of the grokking by connecting the phenomenon with the lottery ticket hypothesis, suggesting that grokking occurs as a result of the discovery of structural exploration (winning lottery tickets).  As a consequence of this, this leads to the phenomenon of accelerated grokking. Not only experiments that show accelerated grokking but also other experiments support our claim that grokking arises from the discovery of structural exploration (winning lottery tickets)
> To clarify this point, we made several modifications. Firstly, we decided to change the title from“GROKKING TICKETS: LOTTERY TICKETS ACCELERATE GROKKING” to “UNDERSTANDING THE INNER-WORKING OF GROKKING THROUGH THE LENS OF LOTTERY TICKETS”.
> Secondly, we summarized the novelty of our research over previous studies on grokking. Specifically, our main claim delves deeper than the analysis of weight norms and weight decay in prior research [Liu 2022, Power 2022], pinpointing structural exploration as the underlying cause. Although the reviewer mentioned that previous research also includes sparsity, our main claim is that for generalization, it's not just sparsity that matters, but the acquisition of good structures is essential. Moreover, by utilizing analysis with lottery tickets instead of neuron pruning, we have conducted experiments in more general tasks and architectures.
>
> ### (4) We added discussion and experiments on the relationship between Grokking Tickets and Lottery tickets in prior research (reviewer KZMf,  DeHf, tMC9).
> We received multiple comments regarding contribution to the research on lottery tickets.
> Firstly, we would like to clarify again that the primary focus of our research is understanding the mechanism of grokking, not proposing new lottery tickets algorithms.
> However, our analysis has yielded several interesting observations for the study of lottery tickets.
> (a) The structures selected as lottery tickets during the transition phase are rapidly converging (refer to Appendix G and H).
> (b) The accelerated generalization with grokking tickets is not merely due to a reduction in parameters but also due to the acquisition of good structures (refer to Appendix I  and J).
> These results seem to be beneficial from the perspective of lottery ticket research, and we have clearly described them in the main text (refer to section 6.2 in the main text).
>
> ### (5)We have revised the text for clarity and ease of understanding (reviewer NUTh, DeHf, tMC9).
> As pointed out in the reviewer's comments, our original writing was unclear and difficult to understand, so we have made revisions for clarity. A significant change is the addition of a section on related work in Section 6. We also have clearly defined Grokking tickets and other comparison models in the abstract and introduction.
>
> ### (6) We added the implementation code can be found at the following link:https://github.com/gouki510/Grokking-Tickets
>
> ### Reference
> Nanda, Neel, et al. "Progress measures for grokking via mechanistic interpretability." arXiv preprint arXiv:2301.05217 (2023).
> Zhong, Ziqian, et al. "The clock and the pizza: Two stories in mechanistic explanation of neural networks." arXiv preprint arXiv:2306.17844 (2023).
> Paganini, Michela, and Jessica Zosa Forde. "Bespoke vs. Pr\^ et-\a-Porter Lottery Tickets: Exploiting Mask Similarity for Trainable Sub-Network Finding." arXiv preprint arXiv:2007.04091 (2020).
>  P. Jaccard. Etude de la distribution florale dans une portion des alpes et du jura. Bulletin de la Societe Vaudoise des Sciences Naturelles, 37:547–579, 01 1901. doi:10.5169/seals-266450.
> Liu, Ziming, Eric J. Michaud, and Max Tegmark. "Omnigrok: Grokking beyond algorithmic data." arXiv preprint arXiv:2210.01117 (2022).
> Merrill, William, Nikolaos Tsilivis, and Aman Shukla. "A Tale of Two Circuits: Grokking as Competition of Sparse and Dense Subnetworks." arXiv preprint arXiv:2303.11873 (2023).
> Power, Alethea, et al. "Grokking: Generalization beyond overfitting on small algorithmic datasets." arXiv preprint arXiv:2201.02177 (2022).

---

> > ### Author Response · Authors · 2023-11-20
> > **General response 3 (Comparison Table with Previous Research)**
> >
> > Thank you again to all reviewers for carefully reading our paper and giving insightful comments.
> > To response to the concern that the comparison with previous research is unclear, we created the comparison table with previous research. We have summarized the aspects in which our research is similar to previous studies and its novelty.
> > |                    | Power 2022, Liu 2022, Google 2023 | Nanda 2023                     | Thilak 2023                                 | Merrill 2023               | Liu 2023                     |
> > |--------------------|-----------------------------------|---------------------------------|---------------------------------------------|----------------------------|-------------------------------|
> > | **Analysis**       | Representation, Weight decay      | Neuron Activity                 | Last layer norm (Slingshot Mechanism)       | Neuron’s Sparsity          | Weight norm, Weight decay     |
> > | **Similarities**   | - Analysis of grokking based on the simplicity of the model. | Each neuron's weight converges to a specific cycle, leading to the discovery of sparse solutions. (See detail in Appendix G) | - Analysis of grokking based on the simplicity of the model, and verification in General Tasks. | - Analysis through network sparsity. - Claim of the emergence of sparse generalization circuits. | - Analysis of grokking based on the simplicity of the model, and verification in General Tasks. |
> > |                    | - Explanation Through the Acquisition of good Representations. |                                |                                             |                            |                               |
> > | **Differences (novelty)**    | - Analysis of what is happening internally in the model from the perspective of Lottery Tickets (LT). | - Analysis of sparsity in weights, not neuron from the perspective of Lottery Tickets (LT) | - Analysis of what is happening internally in the model model from the perspective of Lottery Tickets (LT) | - Analysis of sparsity in weights, not neurons, from the perspective of Lottery Tickets (LT) | - Analysis of what is happening internally in the model model from the perspective of Representation learning |
> > |                    | - Claim that weight decay is being used for pruning weights. | - Analysis in more general tasks, not just tasks like (n-k) parity where sparsity is obvious. | - Analysis of inner-working of models during the transition phase and how models reach generalization solutions. | - Analysis in more general tasks, not just tasks like (n-k) parity where sparsity is obvious. | - Analysis of sparsity, not just weight norms. |
> > |                    | - Analysis of the relationship between representation learning and network structure. | - Analysis of the changes leading to a generalization solution as structural exploration. |                                             | - Analysis of inner-working of models during the transition phase and how models reach generalization solutions. | - Claim that weight decay is being used for pruning weights. |
> > |                |  |  |                                             | Claim that not just sparsity, but the acquisition of good structures is important. | |
> >
> >
> > ## Reference
> > - Power et al. Grokking: Generalization Beyond Overfitting on Small Algorithmic Datasets.
> > - Liu et al. Towards Understanding Grokking: An Effective Theory of Representation Learning.
> > - Nanda et al. Progressive Measures for Grokking via Mechanistic Interpretability.
> > - Pearce et al. (Google, blogpost): Do Machine Learning Models Memorize or Generalize?
> > - Liu et al. Omnigrok: Grokking Beyond Algorithmic Data.
> > - Merrill et al. A Tale of Two Circuits: Grokking as Competition of Sparse and Dense Subnetworks.
> > - Thilak et al. The Slingshot Mechanism: An Empirical Study of Adaptive Optimizers and the Grokking Phenomenon.

---

### Meta-Review · Area_Chair_ZKCy · 2023-12-17

**Metareview:**

This paper presents an attempt to understand the "grokking" phenomenon through the lens of the lottery ticket hypothesis. Three of the five reviewers were unconvinced by the results in the paper and argued for rejection even after the rebuttal. I agreed and opted for rejection. The results are interesting, but I'm confused as to how this provides insight into the grokking phenomenon. The results seem quite similar to the original lottery ticket work (Frankle & Carbin, 2019), where lottery ticket subnetworks learned faster. The authors also deanonymized themselves during the discussion process.

**Justification For Why Not Higher Score:**

The empirical results are neither convincing nor all that significant.

**Justification For Why Not Lower Score:**

N/A

---

### Decision · Program_Chairs · 2024-01-16

Reject